# Research on risk management incentive strategy based on the green financial ecosystem

**ZhongPing Cui[1], Shuang Lu[1]\*, JinRong Liu[2]**

**1** School of Economics, Shenyang University of Technology, Shenyang, China, **2** Liaoning Zhixing Valley Technology Co., LTD, Shenyang, China

\* 201003@smail.sut.edu.cn

**Data Availability Statement:** All relevant data are within the paper.

**Funding:** The author(s) received no specific funding for this work.

## Abstract

Taking the green financial ecosystem composed of innovators, green financial institutions and regulators as the object of research, it explores the issue of how to improve the level of efforts of the three types of subjects and the benefits of risk management in the green financial ecosystem. The optimal level of effort, optimal level of return, and optimal level of return on risk management of green financial ecosystems for innovators, green financial institutions, and regulators under the three modes of No-incentive Contract, Cost-sharing Contract, and Synergistic Cooperation Contract are investigated and analyzed respectively, and verified by numerical simulation analysis. The results show: (1) Compared to the No-incentive Contract, the Cost-sharing Contract and the Synergy Cooperation Contract generate more significant incentives, and returns increase over time in both models. (2) The effort level of the participating subjects under the Synergistic Cooperation Contract is the highest, which can realize the Pareto optimization of the participating subjects and the green financial ecosystem at the same time. The study's findings contribute to a deeper understanding of cooperation among innovators, green financial institutions and regulators in facilitating risk management in green financial ecosystems and provide a realistic reference for risk managers in green financial ecosystems.

## Introduction

Green finance is one of the essential components of the international policy of "Carbon Peak and Carbon Neutrality", playing a positive role in guiding the rational allocation of resources, promoting the transfer of resources for green and low-carbon projects, and facilitating the ecologization of traditional industries and the development of new types of green and ecological industries. The development of green finance is a crucial way to promote the construction of an ecological civilization, win the battle against pollution and achieve the goal of "Carbon Peak and Carbon Neutrality". The global goal of "Carbon Peak and Carbon Neutrality" aims to reshape countries' energy structure and industrial structure worldwide through a series of green transformation projects involving many aspects of economic and social development

**Competing interests:** The authors have declared that no competing interests exist.

[1]. In the context of this objective, the importance of green finance for economic development has become more prominent. Green finance can help countries worldwide achieve the goal of low carbon emissions and contribute to the realization of green and sustainable growth of the global economy, which is of great significance in addressing global climate and environmental issues. In order to cope with the challenges of global climate change, countries around the world have successively set up "Carbon Peak and Carbon Neutrality" goals in line with the national conditions of each country [2], and in this process, green finance will play an essential leading role, which can guide the flow of funds, and then change the allocation of resources, and ultimately realize the goal of driving the industrial structure to the direction of green environmental protection and low-carbon upgrading.

In recent years, countries worldwide have been paying more and more attention to the development of green finance, giving support and encouragement in terms of guidelines and policies. And green finance has also received attention. Green financial risks are quite different from traditional risks. On the one hand, green project risks have great uncertainty, and it is difficult to identify them. On the other hand, it is difficult to manage the risks affecting the implementation of green finance. Green financial risk management can start by reducing the risk of green investment projects, green supply chains, green strategies, etc., to strengthen the management of green risk, curbing the risk from the source, and monitoring the risk changes promptly so that the risk is in a controllable range, which is a significant development goal of the green financial industry [3]. In the context of the dual-carbon strategy, green finance should positively respond, constantly improve its essential capacity building, and form a rich and systematic green financial risk management system to effectively enhance green financial risk resistance and promote the healthy development of green finance.

This paper studies the risk management of green financial ecosystems composed of innovators, green financial institutions and regulators with the help of differential game models, focusing on the incentive effects under different contracts. Calculate the optimal strategies, optimal returns and optimal returns of green financial ecosystems of the participating subjects under the three modes of no-incentive contract, cost-sharing contract and synergistic contract, and then obtain the optimal strategies of innovators, green financial institutions and regulators in the process of cooperation and further analyze the key factors affecting the returns of green financial ecosystems, to provide decision-making support for the incentives and benefit distribution of risk management of green financial ecosystems.

## Literature review

### Green finance

Green finance refers to supporting and implementing environmental protection and improvement, active response to climate change. It enhances the efficiency of resource utilization of economic industries and activities for environmental protection, energy saving, clean energy, green transportation, green building and other different areas and related projects to provide financial products and services to match their needs [4]. With the rise of the concept and practice of green finance, scholars have carried out multi-dimensional research, mainly focusing on the following aspects.

The first is studies on the theoretical mechanisms of the green financial system. Ma [5] analyzed the necessity of creating a green financial system and, through the combing of foreign literature, proposed to build China's green financial system from the perspectives of institutional construction, policy support and legal protection. Mohd et al. [6] considered green finance as a financial system that uses the market as a means of resource allocation and finance as a tool to improve the ecological environment. Falcone [7] believes that green finance through the

 

investment of funds to the ecological environment as a goal to achieve the goal of relying on financial means to protect the environment and save resources. Deng et al. [8] argued that the precondition for the development of green finance is the sound management of commercial banks, based on which the green financial development index system was constructed from green business, green operation and green strategy.

The second is a study on the dynamics of green financial development. Guo et al. [9] empirically analyze the impact of green finance on renewable energy development with the help of the mediation effect model and China's provincial panel data from 2007–2019 and find that the two paths of green finance affecting renewable energy development are clean-biased technological progress path and financial constraints path and that there are significant differences in the driving roles of green finance development and government policies at different stages. Yang et al. [10] argued that green finance has become the main driver of green development, in which green innovation is the critical way, and with the help of a fixed-effects model, the hypotheses were empirically tested with the data of 30 provinces in China from 2008–2019, and the results of the study showed that the role of green finance in promoting green innovation has been enhanced in all the regions through strict environmental regulation. By constructing a comprehensive index system, Han [11] found that green credit, green securities and green insurance all help to improve the efficiency of green technology innovation. Guo [12] suggested that green financial innovation can help realize the goal of "carbon peak and carbon neutral".

The third is that some scholars have analyzed and explored the behavioral decision-making of green financial institutions with the help of constructing a game model between the government, banks and enterprises. Li et al. [13] In order to study the development mechanism of renewable energy + storage cooperation under government participation, this paper constructs a tripartite evolutionary game model between the government, renewable energy generators and storage service providers based on the data on renewable energy + storage projects in a Chinese province. Xu et al. [14] from the game between polluting enterprises, policy banks and regulators, found that reducing the cost of penalties imposed by regulators on enterprises and improving the economic efficiency of banks in implementing green financial policies will help the implementation of green financial policies. Chen et al. [15] analyzed the behavior of enterprises and banks in the state of government and anarchy with the help of a game model and found that the government has a positive role in promoting the development of green finance. Zhang et al. [16] analyzed the impact of government subsidies on the overall system benefits under different contractual modes by constructing a differential game model and found that the government bears part of the green costs through taxes or financial subsidies, which contributes to Pareto improvement.

### Green finance risk management

Under the new environment of global active implementation of carbon emission reduction and vigorous development of green economy, the "double carbon" goal is becoming the wind vane for countries to optimize industrial structure, develop green economy and achieve high-quality development. The development of green finance has also ushered in new opportunities. Building an effective and comprehensive green financial risk management system, improving the ability to prevent and resolve green financial risks, and escorting the sustained and healthy development of green finance are also challenges in the development of green finance [17]. Scholars have carried out many studies on green financial risks.

The first is a study of theoretical mechanisms relating to financial risk management. Li et al. [18] argued that green finance has the superimposed risk of finance and the environment, and

in this regard, analyzed the formation mechanism and coping strategy of green finance risk. Luo [19] believes that in order to ensure the sound development of enterprises, the financial management department should correctly recognize the types of financial risks in the investment and financing links and take targeted preventive and control measures. VISWA-NATHAN et al. [20] study risk management in financial institutions with the help of data on interest rate hedging and foreign exchange risk. Li [21] believes that the establishment of a comprehensive, full-process financial risk management system can effectively resolve financial risks. Chen et al. [22] believe intelligence services provide new ideas and ways for green financial risk management. Applying intelligence services to green financial risk management will help the sustainable development of green finance and enrich the theoretical system of intelligence science.

The second is a study on financial risk assessment methodologies. Cerchiello et al. [23] argued that a critical area of financial risk management is systematic risk modeling, and in this regard, the two scholars developed a novel systematic risk model and estimated a systematic risk model using two different data sources (financial markets and financial tweets), and proposed a Bayesian approach to combine them. Su et al. [24] proposed the blockchain-based corporate financial risk prevention and early warning system. Zhao [25] further explores the influence path of Internet finance on the risk prevention and management of commercial banks, with the help of a back-propagation neural network optimization algorithm to predict the value of risk and empirical research and analysis of the changes in the level of risk of commercial banks in the Internet environment. Tao et al. [26] constructed the estimation index system of green financial risk with the help of diamond model and network analysis. Hu et al. [27] constructed a green financial risk evaluation index system for the basic situation of China's green finance, starting from the supply and demand links of governmental departments, green financial market and green funds, and calculated the weight of each index in China's green financial risk from each dimension with the help of hierarchical analysis. Zhao [28] studied the supply chain green financial risk management and green financial risk management evaluation indexes through the BP neural network method, which showed that the evaluation results have high scientific validity.

## Problem description and model assumptions

### Problem description

In the context of green finance, the study of cooperation among multiple actors within the green financial ecosystem to improve the risk management of the green financial ecosystem focuses on the impact of different cooperation models and incentive strategies on the participating actors and the green financial ecosystem. The conceptual model constructed in this paper is shown in Figs 1 and 2. The innovator improves the maturity level of green products mainly by investing technological innovation resources in research, development, and design. Green financial institutions are mainly through the production of green financial products to realize the input of green financial products to improve the level of risk management, and specific input areas can be divided into green loans, green bonds and so on. Regulators guide and regulate policies to incentivize the market and innovative parties to invest in the level of effort by establishing appropriate incentives and disincentives, such as the provision of loan subsidies, green subsidies and administrative regulation. The above human, technical, financial, information, and other related resource costs incurred by innovators, green financial institutions, and regulators in carrying out these behaviors are collectively referred to as their respective efforts towards risk management in the green financial ecosystem and are measured in terms of the level of effort.

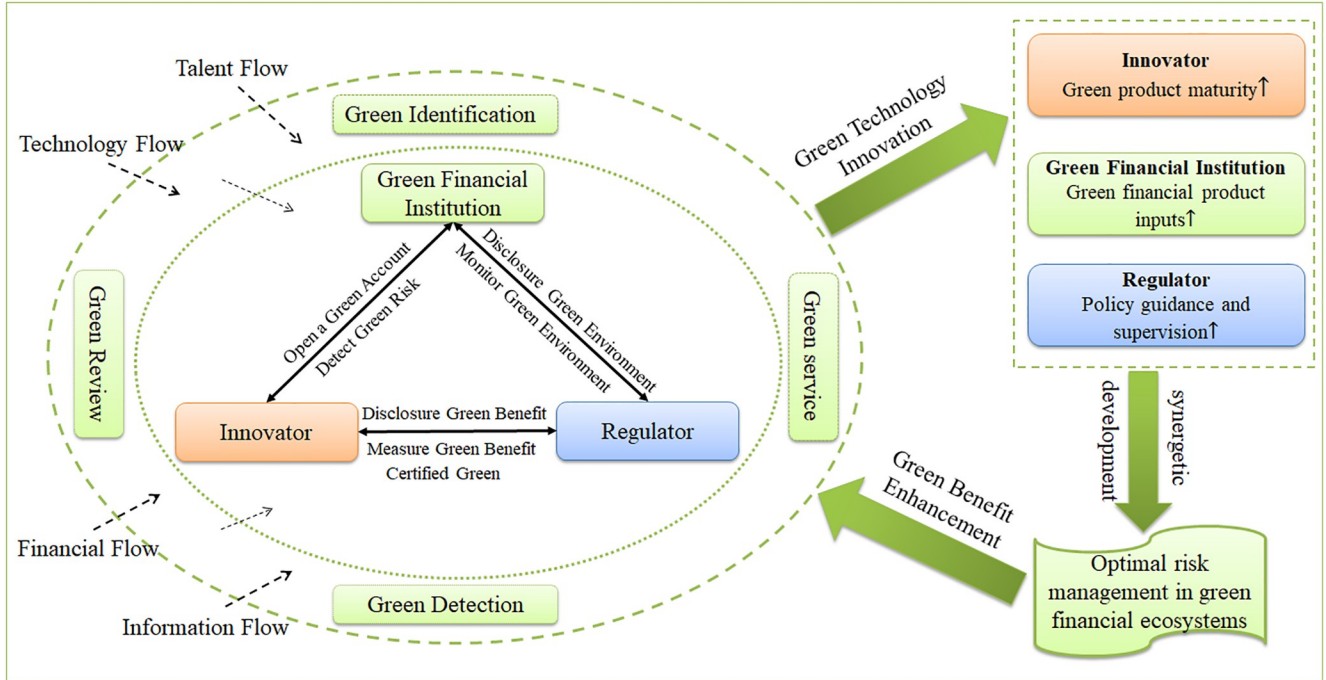

**Fig 1. Operational mechanism chart of risk management in green financial ecosystem.**

## Model assumption

Based on the background of the problem, the following hypotheses are made in this paper:

**Hypothesis 1:** The level of effort invested in risk management of the green financial ecosystem by each of the innovators, green financial institutions, and regulators at the moment t is respectively $E_I(t)$, $E_F(t)$ and $E_R(t)$, where $t(t \in [0, \infty))$ is the time variable. The input costs of innovators, green financial institutions, and regulators are related to their respective levels of effort invested in green financial risk management, with input costs being a convex function of the level of effort, i.e., the higher the level of effort invested, the higher the cost of the input. Input costs are considered in two main ways: Human resource costs

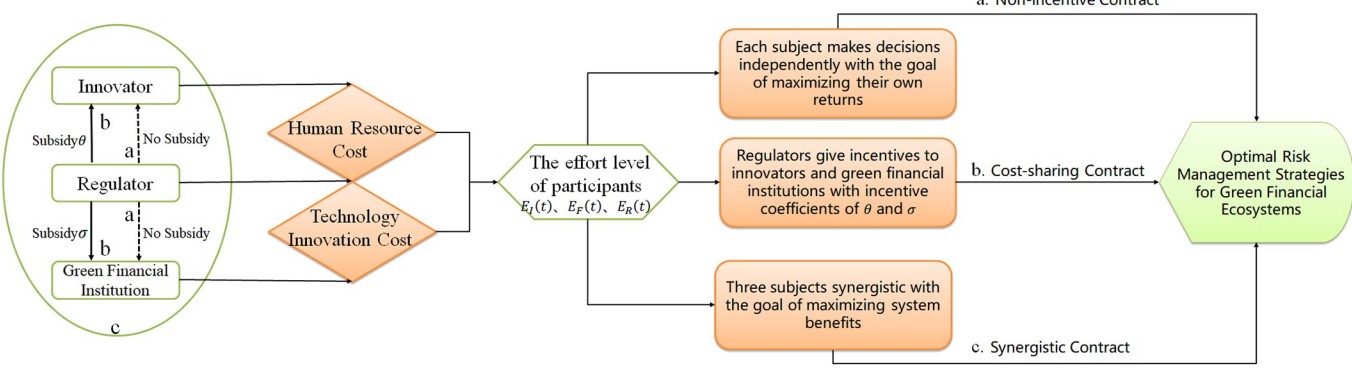

**Fig 2. Construction and solution flow chart of three contract modes.**

$\frac{\mu_i}{2}E_i^2(t)(i = I, F, R)$, and Technological innovation costs $\frac{1}{k_i}E_i^2(t)(i = I, F, R)$.

$$C_I(t) = \frac{1}{k_I}E_I^2(t)$$
$$C_F(t) = (\frac{\mu_F}{2} + \frac{1}{k_F})E_F^2(t) \tag{1}$$
$$C_R(t) = \frac{\mu_R}{2}E_R^2(t)$$

Where $\mu_i$ ($i = I, F, R$) denotes the human resource cost factor, $K_i$ ($i = I, F, R$) denotes the coefficient of familiarity of technological innovation, the higher the coefficient of maturity of technological innovation, the lower the cost paid by the process of risk management enhancement in the green financial ecosystem. $C_I(t)$, $C_F(t)$ and $C_R(t)$ represent the cost of risk management inputs to the green financial ecosystem by innovators, green financial institutions, and regulators at the moment $t$ respectively.

**Hypothesis 2:** The enhancement of green financial ecosystem risk management is a dynamic process which requires continuous cooperation and innovation of the participating entities to achieve the optimal green financial ecosystem risk management level ultimately. In this paper, the level of risk management in green financial ecosystems is regarded as a state variable, which is influenced by the level of effort and technological innovation of each participant. Assume that the level of risk management in the green financial ecosystem at the moment $t$ is $G(t)$, the process of improving the level of risk management in the green financial ecosystem satisfies the following Eq.

$$G'(t) = \lambda_I E_I(t) + \lambda_F E_F(t) + \lambda_R E_R(t) - \delta G(t) \tag{2}$$

Where the initial level of green financial ecosystem risk management is $G(0) = G_0 \geq 0$, $\lambda_i$ ($i = I, F, R$) $> 0$ denotes the coefficients of the impact of the level of effort invested by the three agents on the risk management of the green financial ecosystem, respectively. $\delta > 0$ represents the natural decay rate of the risk management level of the green financial ecosystem over time, subject to uncertainty factors.

**Hypothesis 3:** The total green finance ecosystem benefit $\pi(t)$ at time $t$ is:

$$\pi(t) = \varepsilon_I E_I(t) + \varepsilon_F E_F(t) + \varepsilon_R E_R(t) + \eta G(t) \tag{3}$$

Where $\varepsilon_i(i = I, F, R)$ is the extent to which the level of effort affects the level of risk management in the green financial ecosystem, i.e., the marginal benefit coefficient, and $\eta$ is the extent to which the level of technological innovation affects the total benefit, $\eta > 0$.

**Hypothesis 4:** The total benefits of the green finance ecosystem are distributed among three types of agents, with a coefficient of distribution of benefits among innovators, green financial institutions, and regulators: $\alpha, \beta, 1 - \alpha - \beta$ ($0 < \alpha, \beta, 1 - \alpha - \beta < 1$), regulators sharing the costs of research and development for innovators and green financial institutions, the sharing ratios are $\theta$ and $\sigma$ ($0 \leq \theta, \leq 1$), consider this ratio as an incentive factor for regulators to incentivize innovators and green financial institutions. It is assumed that the innovator, the green financial institution, and the regulator are perfectly informed and that the same positive discount rate $\rho$ exists.

The objective function of the innovator is:

$$\max_{E_I(t)} J_I = \int_0^\infty e^{-\rho t}[\alpha\pi(t) - (1-\theta(t))\frac{1}{k_I}E_I^2(t)]dt \tag{4}$$

The objective function of green financial institutions is:

$$\max_{E_F(t)} J_F = \int_0^\infty e^{-\rho t}[\beta\pi(t) - (1-\sigma(t))(\frac{\mu_F}{2}+\frac{1}{k_F})E_F^2(t)]dt \tag{5}$$

The objective function of the regulator is:

$$\max_{E_R(t)} J_R = \int_0^\infty e^{-\rho t}[(1-\alpha-\beta)\pi(t) - \frac{\mu_R}{2}E_R^2(t) - \theta(t)\frac{1}{k_I}E_I^2(t) - \sigma(t)(\frac{\mu_F}{2}+\frac{1}{k_F})E_F^2(t)]dt \tag{6}$$

This model contains control variables $E_I$(t), $E_F$(t), $E_R$(t), $\theta$ and $\sigma$, and the state variable $G(t)$, Since the model cannot be solved under dynamic parameter conditions, it is assumed that all parameters in this paper are time-independent positive constants, and the time variable $t$ will be omitted later for convenience. At any point in time in the infinite time zone, the innovator, the green financial institution and the regulator face the same game, so referring to Fu et al. [29], the strategy is restricted to a static strategy and the corresponding static feedback equilibrium solution is obtained.

## Modeling and solving

### Non-incentive contract

In a Non-incentive Contract, the regulator does not provide any incentives for innovators and green financial institutions, i.e., $\theta = 0$, $\sigma = 0$. The three types of subjects are equal and independent of each other, all to achieve their revenue maximization as the goal, each will make the most advantageous decision for their own, the optimal combination of strategies at this time to achieve the Nash equilibrium state. Any subject who unilaterally changes his strategy under this combination of strategies (other subjects' strategies remain unchanged) will not increase his own gain.

The benefit functions for innovators, green financial institutions and regulators are $V_I(G)$, $V_F(G)$ and $V_R(G)$, which are continuously bounded and differentiable, and all satisfy the following *HJB* equation when $G \geq 0$:

$$\rho V_I(G) = \max_{E_I \geq 0}[\alpha\pi(t) - \frac{\mu_I}{2}E_I^2 + V_I'(G)(\lambda_I E_I + \lambda_F E_F + \lambda_R E_R - \delta G)] \tag{7}$$

$$\rho V_F(G) = \max_{E_F \geq 0}[\beta\pi(t) - (\frac{\mu_F}{2}+\frac{1}{k_F})E_F^2 + V_F'(G)(\lambda_I E_I + \lambda_F E_F + \lambda_R E_R - \delta G)] \tag{8}$$

$$\rho V_R(G) = \max_{E_R \geq 0}[(1-\alpha-\beta)\pi(t) - \frac{\mu_R}{2}E_R^2 + V_R'(G)(\lambda_I E_I + \lambda_F E_F + \lambda_R E_R - \delta G)] \tag{9}$$

The right end of Eqs (7)–(9) are used to obtain the first partial derivative of $E_I$, $E_F$, and $E_R$ respectively, and make it equal to zero. The solution is:

$$E_I = \frac{k_I(\alpha\varepsilon_I + \lambda_I V_I'(G))}{2}$$

$$E_F = \frac{k_F(\beta\varepsilon_F + \lambda_F V_F'(G))}{2 + k_F\mu_F}, \quad E_R = \frac{(1 - \alpha - \beta)\varepsilon_R + \lambda_R V_R'(G)}{\mu_R} \tag{10}$$

Substitute the Eq (10) into Eqs (7)–(9) to simplify:

$$\rho V_I(G) = (\alpha\eta - \delta V_I'(G))G + \frac{\alpha^2 k_I \varepsilon_I^2 + k_I \lambda_I V_I'(G)(2\alpha\varepsilon_I + V_I'(G)\lambda_I)}{4}$$

$$+ \frac{k_F(\beta\varepsilon_F + V_F'(G)\lambda_F)(\alpha\varepsilon_F + V_I'(G)\lambda_F)}{2 + k_F\mu_F} + \frac{((1 - \alpha - \beta)\varepsilon_R + V_R'(G)\lambda_R)(\alpha\varepsilon_R + V_I'(G)\lambda_R)}{\mu_R} \tag{11}$$

$$\rho V_F(G) = (\beta\eta - \delta V_F'(G))G + \frac{(\alpha\varepsilon_I + V_I'(G)\lambda_I)(\beta k_I\varepsilon_I + k_I\lambda_I V_F'(G))}{2}$$

$$+ \frac{k_F(\beta\varepsilon_F + V_F'(G)\lambda_F)^2}{2(2 + k_F\mu_F)} + \frac{((1 - \alpha - \beta)\varepsilon_R + V_R'(G)\lambda_R)(\beta\varepsilon_R + V_F'(G)\lambda_R)}{\mu_R} \tag{12}$$

$$\rho V_R(G) = ((1 - \alpha - \beta)\eta - \delta V_R'(G))G + \frac{((1 - \alpha - \beta)k_I\varepsilon_I + V_R'(G)k_I\lambda_I)(\alpha\varepsilon_I + V_I'(G)\lambda_I)}{2}$$

$$+ \frac{((1 - \alpha - \beta)k_F\varepsilon_F + V_R'(G)k_F\lambda_F)(\beta\varepsilon_F + V_F'(G)\lambda_F)}{2 + k_F\mu_F} \tag{13}$$

$$+ \frac{((1 - \alpha - \beta)\varepsilon_R + V_R'(G)\lambda_R)(2((1 - \alpha - \beta)\varepsilon_R + V_R'(G)\lambda_R) + ((1 - \alpha - \beta)\varepsilon_R + V_R'(G)\lambda_R))}{2\mu_R}$$

From the form of Eqs (11)–(13), it can be seen that the one-variable function $G$ as the independent variable is the solution of the *HJB* equation, so:

$$V_I(G) = f_1 G + f_2, \quad V_F(G) = g_1 G + g_2, \quad V_R(G) = h_1 G + h_2 \tag{14}$$

Where $f_1, f_2, g_1, g_2, h_1, h_2$ are the constants to be solved, we get:

$$V_I'(G) = \frac{dV_I(G)}{dG} = f_1, \quad V_F'(G) = \frac{dV_F(G)}{dG} = g_1, \quad V_R'(G) = \frac{dV_R(G)}{dG} = h_1 \tag{15}$$

Substitute Eqs (14) and (15) into Eqs (11)–(13), we get:

$$
\begin{aligned}
\rho(f_1 G + f_2) &= (\alpha\eta - f_1\delta)G + \frac{\alpha^2 k_I \varepsilon_I^2 + k_I \lambda_I f_1(2\alpha\varepsilon_I + f_1\lambda_I)}{4} \\
&+ \frac{k_F(\alpha\varepsilon_F + f_1\lambda_F)(\beta\varepsilon_F + g_1\lambda_F)}{2 + k_F\mu_F} + \frac{(\alpha\varepsilon_R + f_1\lambda_R)((1-\alpha-\beta)\varepsilon_R + h_1\lambda_R)}{\mu_R}
\end{aligned}
\tag{16}
$$

$$
\begin{aligned}
\rho(g_1 G + g_2) &= (\beta\eta - \delta g_1)G + \frac{(\alpha\varepsilon_I + f_1\lambda_I)(\beta k_I \varepsilon_I + g_1 k_I \lambda_I)}{2} + \frac{k_F(\beta\varepsilon_F + g_1\lambda_F)^2}{2(2 + k_F\mu_F)} \\
&+ \frac{(\beta\varepsilon_R + g_1\lambda_R)((1-\alpha-\beta)\varepsilon_R + h_1\lambda_R)}{\mu_R}
\end{aligned}
\tag{17}
$$

$$
\begin{aligned}
\rho(h_1 G + h_2) &= ((1-\alpha-\beta)\eta - h_1\delta)G + \frac{((1-\alpha-\beta)k_I\varepsilon_I + h_1 k_I \lambda_I)(\alpha\varepsilon_I + f_1\lambda_I)}{2} \\
&+ \frac{k_F\varepsilon_F((1-\alpha-\beta) + h_1)(\beta\varepsilon_F + g_1\lambda_F)}{2 + k_F\mu_F} \\
&+ \frac{((1-\alpha-\beta)\varepsilon_R + h_1\lambda_R)(2((1-\alpha-\beta)\varepsilon_R + h_1\lambda_R) + ((1-\alpha-\beta)\varepsilon_R + h_1\lambda_R))}{2\mu_R}
\end{aligned}
\tag{18}
$$

It was previously assumed that $V_I(G)$, $V_F(G)$ and $V_R(G)$ satisfy the conditions at $G \geq 0$, so $f_1, f_2, g_1, g_2, h_1, h_2$ are obtained as:

$$
f_1 = \frac{\alpha\eta}{\rho + \delta},\ g_1 = \frac{\beta\eta}{\rho + \delta}, h_1 = \frac{(1-\alpha-\beta)\eta}{\rho + \delta}
\tag{19}
$$

$$
f_2 = \frac{\alpha^2 k_I((\delta+\rho)\varepsilon_I + \eta\lambda_I)^2}{4\rho(\delta+\rho)^2} + \frac{\alpha\beta k_F((\delta+\rho)\varepsilon_F + \eta\lambda_F)^2}{\rho(\delta+\rho)^2(2 + k_F\mu_F)} + \frac{\alpha(1-\alpha-\beta)((\delta+\rho)\varepsilon_R + \eta\lambda_R)^2}{\mu_R\rho(\delta+\rho)^2}
\tag{20}
$$

$$
g_2 = \frac{\alpha\beta k_I((\delta+\rho)\varepsilon_I + \eta\lambda_I)^2}{2\rho(\delta+\rho)^2} + \frac{\beta^2 k_F((\delta+\rho)\varepsilon_F + \eta\lambda_F)^2}{2\rho(\delta+\rho)^2(2 + k_F\mu_F)} + \frac{\beta(1-\alpha-\beta)((\delta+\rho)\varepsilon_R + \eta\lambda_R)^2}{\mu_R\rho(\delta+\rho)^2}
\tag{21}
$$

$$
h_2 = \frac{\alpha k_I(1-\alpha-\beta)((\delta+\rho)\varepsilon_I + \eta\lambda_I)^2}{2\rho(\delta+\rho)^2} + \frac{\beta k_F(1-\alpha-\beta)((\delta+\rho)\varepsilon_F + \eta\lambda_F)^2}{\rho(\delta+\rho)^2(2 + k_F\mu_F)} + \frac{(1-\alpha-\beta)^2((\delta+\rho)\varepsilon_R + \eta\lambda_R)^2}{2\mu_R\rho(\delta+\rho)^2}
\tag{22}
$$

Substitute $f_1$, $g_1$, $h_1$ into Eq (10) to obtain the maximum effort level of the innovative party, green financial institution and regulatory party as follows:

$$
E_{I1}^* = \frac{\alpha k_I(\varepsilon_I(\delta+\rho) + \eta\lambda_I)}{2(\delta+\rho)},\ E_{F1}^* = \frac{\beta k_F(\varepsilon_F(\delta+\rho) + \eta\lambda_F)}{(\delta+\rho)(2 + k_F\mu_F)}
$$

$$
E_{R1}^* = \frac{(1-\alpha-\beta)(\varepsilon_R(\delta+\rho) + \eta\lambda_R)}{\mu_R(\delta+\rho)}
\tag{23}
$$

Substitute $f_1, f_2, g_1, g_2, h_1, h_2$ into Eq (14), the optimal income function of the innovative party, green financial institution and the regulatory party can be obtained as follows:

$$
\begin{aligned}
V_{I1}(G)^* &= \frac{\alpha\eta}{(\delta+\rho)}G + \frac{\alpha^2 k_I((\delta+\rho)\varepsilon_I + \eta\lambda_I)^2}{4\rho(\delta+\rho)^2} + \frac{\alpha\beta k_F((\delta+\rho)\varepsilon_F + \eta\lambda_F)^2}{\rho(2+k_F\mu_F)(\delta+\rho)^2} \\
&+ \frac{\alpha(1-\alpha-\beta)((\delta+\rho)\varepsilon_R + \eta\lambda_R)^2}{\rho\mu_R(\delta+\rho)^2}
\end{aligned}
\tag{24}
$$

$$
\begin{aligned}
V_{F1}(G)^* &= \frac{\beta\eta}{(\delta+\rho)}G + \frac{\alpha\beta k_I((\delta+\rho)\varepsilon_I + \eta\lambda_I)^2}{2\rho(\delta+\rho)^2} + \frac{\beta^2 k_F((\delta+\rho)\varepsilon_F + \eta\lambda_F)^2}{2\rho(\delta+\rho)^2(2+k_F\mu_F)} \\
&+ \frac{\beta(1-\alpha-\beta)((\delta+\rho)\varepsilon_R + \eta\lambda_R)^2}{2\rho\mu_R(\delta+\rho)^2}
\end{aligned}
\tag{25}
$$

$$
\begin{aligned}
V_{R1}(G)^* &= \frac{(1-\alpha-\beta)\eta}{(\delta+\rho)}G + \frac{(1-\alpha-\beta)}{2\rho(\delta+\rho)^2}(\alpha k_I((\delta+\rho)\varepsilon_I + \eta\lambda_I)^2 \\
&+ \frac{2\beta k_F((\delta+\rho)\varepsilon_F + \eta\lambda_F)^2}{(2+k_F\mu_F)} + \frac{(1-\alpha-\beta)((\delta+\rho)\varepsilon_R + \eta\lambda_R)^2}{\mu_R})
\end{aligned}
\tag{26}
$$

The benefits of the green financial ecosystem are:

$$
\begin{aligned}
V(G)^* &= \frac{\eta}{(\delta+\rho)}G + \frac{1}{4\rho(\delta+\rho)^2}(((\delta+\rho)\varepsilon_I + \eta\lambda_I)^2(\alpha k_I(2-\alpha)) \\
&+ \frac{2\beta k_F((\beta+4\rho(\delta+\rho)^2) + 4\alpha\beta k_F)((\delta+\rho)\varepsilon_F + \eta\lambda_F)^2}{(2+k_F\mu_F)} \\
&+ \frac{4(1-\alpha-\beta)(\rho(\delta+\rho)^2(1-\alpha)+\alpha)((\delta+\rho)\varepsilon_R + \eta\lambda_R)^2}{\mu_R})
\end{aligned}
\tag{27}
$$

## Cost-sharing contract

In the Cost-sharing Contract, the regulator gives cost subsidies $\theta$ and $\sigma$ ($0 \leq \sigma, \theta \leq 1$) to the innovator and the green financial institution, respectively, to improve innovation and green financial institutions research and development enthusiasm. The regulator first determines the cost-subsidy ratio and the level of effort, and innovators and green financial institutions' observe the regulator's decisions and make the appropriate follow-through decisions to ensure that they maximize their returns. Regulators should be able to anticipate the response strategies of innovators and green financial institutions before making final decisions. Therefore, the Stackelberg equilibrium state in this state is solved by backward induction. Within this framework, the leader acts first to set a strategy, and later, the followers will decide their optimal response based on the leader's strategy.

It is assumed that the optimal income functions $V_I(G)$, $V_F(G)$ and $V_R(G)$ of the three types of agents are continuously bounded and differentiable, And when $G \geq 0$, the *HJB* equation is satisfied. The *HJB* equations for innovative parties and green financial institutions are as

follows:

$$\rho V_I(G) = \max_{E_I \geq 0}\left[\frac{(\theta - 1)E_I^2}{k_I} + \alpha(E_F\boldsymbol{\varepsilon}_F + E_I\boldsymbol{\varepsilon}_I + E_R\boldsymbol{\varepsilon}_R + \eta G) + V_I'(G)(E_F\lambda_F + E_I\lambda_I + E_R\lambda_R - \delta G)\right] \quad (28)$$

$$\rho V_F(G) = \max_{E_F \geq 0}\left[\beta(E_F\boldsymbol{\varepsilon}_F + E_I\boldsymbol{\varepsilon}_I + E_R\boldsymbol{\varepsilon}_R + \eta G) + V_F'(G)(E_F\lambda_F + E_i\lambda_i + E_R\lambda_R - \delta G) + (\sigma - 1)(\frac{E_F^2}{k_F} + \frac{\mu_F E_F^2}{2})\right] \quad (29)$$

Take the first partial derivative of the right end of Eqs (28) and (29), respectively concerning $E_I$ and $E_F$, and make it equal to zero, we get:

$$E_I = \frac{k_I(\alpha\boldsymbol{\varepsilon}_I + V_I'(G)\lambda_I)}{2(1-\theta)}, \; E_F = \frac{k_F(\beta\boldsymbol{\varepsilon}_F + V_F'(G)\lambda_F)}{(1-\sigma)(2+k_F\mu_F)} \quad (30)$$

The regulator decides the level of its efforts and the proportion of costs subsidized to other subjects based on the decisions of innovators and green financial institutions, and its *HJB* equation can be expressed as follows:

$$\rho V_R(G) = \max_{E_R \geq 0}\left\{-\frac{\theta E_I^2}{k_I} + (1 - \alpha - \beta)(E_F\boldsymbol{\varepsilon}_F + E_I\boldsymbol{\varepsilon}_I + E_R\boldsymbol{\varepsilon}_R + \eta G)\right.$$
$$\left. + V_R'(G)(E_F\lambda_F + E_I\lambda_I + E_R\lambda_R - \delta G) - \sigma(\frac{E_F^2}{k_F} + \frac{\mu_F E_F^2}{2}) - \frac{\mu_R E_R^2}{2}\right\} \quad (31)$$

Substitute Eq (30) into Eq (31), and solve the right-hand part to maximize it under the condition that Eq (31) takes the first partial derivative concerning $E_R$, $\theta$ and $\sigma$ equal to zero and obtains:

$$E_R = \frac{(1 - \alpha - \beta)\boldsymbol{\varepsilon}_R + V_F'(G)\lambda_R}{\mu_R} \quad (32)$$

$$\theta = \frac{(2 - 3\alpha - 2\beta)\boldsymbol{\varepsilon}_I + (2V_R'(G) - V_I'(G))\lambda_I}{(2 - \alpha - 2\beta)\boldsymbol{\varepsilon}_I + (V_I'(G) + 2V_R'(G))\lambda_I} \quad (33)$$

$$\sigma = \frac{(2 - 2\alpha - 3\beta)\boldsymbol{\varepsilon}_F + (2V_R'(G) - V_F'(G))\lambda_F}{(2 - 2\alpha - \beta)\boldsymbol{\varepsilon}_F + (V_F'(G) + 2V_R'(G))\lambda_F} \quad (34)$$

Substitute the Eq (30) and the Eqs (32)–(34) into the *HJB* equation simplifies to:

$$\rho V_I(G) = \frac{\alpha\eta(2 + k_F\mu_F) - 4\delta V_I'(G)}{4}G + \frac{k_F\lambda_F V_I'(G)(\varepsilon_F(2 - 2\alpha - \beta) + \lambda_F(V_F'(G) + 2V_R'(G)))}{2(2 + k_F\mu_F)}$$

$$+ \frac{(\alpha k_I\varepsilon_I + k_I\lambda_I V_I'(G))((2 - \alpha - 2\beta)\varepsilon_I + \lambda_I V_I'(G) + 2\lambda_I V_R'(G))}{8}$$

$$+ \frac{\alpha k_F\varepsilon_F((2 - 2\alpha - \beta)\varepsilon_F + \lambda_F V_F'(G) + 2\lambda_F V_R'(G))}{8} + \frac{(\alpha\varepsilon_R + \lambda_R V_I'(G))((1 - \alpha - \beta)\varepsilon_R + \lambda_R V_R'(G))}{\mu_R}$$

(35)

$$\rho V_F(G) = (\beta\eta - \delta V_F'(G))G + \frac{(\beta k_I\varepsilon_I + k_I\lambda_I V_F'(G))((2 - \alpha - 2\beta)\varepsilon_I + \lambda_I V_I'(G) + 2\lambda_I V_R'(G))}{4}$$

$$+ \frac{V_F'(G)(k_F V_F'(G)\lambda_F^2 + (2(1 - \alpha)k_F\lambda_F\varepsilon_F^2) + 2k_F\lambda_F V_R'(G)(V_F'(G)\lambda_F + \beta\varepsilon_F^2) + \beta k_F\varepsilon_F^2(2 - 2\alpha - \beta)}{4(2 + k_F\mu_F)}$$

$$+ \frac{(\beta\varepsilon_R + \lambda_R V_F'(G))(\varepsilon_R(1 - \alpha - \beta) + \lambda_R V_R'(G))}{\mu_R}$$

(36)

$$\rho V_R(G) = (\eta + \delta V_R'(G))G + \frac{2\eta G}{2 + k_F\mu_F} + \frac{((1 - \alpha - \beta)\varepsilon_R + V_R\lambda_R)^2}{2\mu_R}$$

$$+ \frac{\varepsilon_I k_I\lambda_I(2 - \alpha - 2\beta)V_R'(G)}{4} + \frac{k_I((2 - \alpha - 2\beta)\varepsilon_I + \lambda_I V_I'(G))^2}{16} + \frac{V_R'(G)k_I\lambda_I^2(1 + V_R'(G))}{4}$$

$$+ \frac{2k_F((2 - 2\alpha - \beta)\varepsilon_F - \lambda_F V_F'(G))^2 + 8k_F\lambda_F^2(V_R'(G))^2}{16(2 + k_F\mu_F)}$$

$$+ \frac{4V_R'(G) + 8V_R'(G)k_F\lambda_F(2 - 2\alpha - \beta)\varepsilon_F + 8k_F\lambda_F^2 V_R'(G)V_F'(G)}{16(2 + k_F\mu_F)}$$

(37)

From the structure of Eqs (35)–(37), it can be seen that the unary one-time function *G* as the independent variable is the solution of the *HJB* equation, so:

$$V_I(G) = f_1 G + f_2, \ V_F(G) = g_1 G + g_2, \ V_R(R) = h_1 G + h_2 \tag{38}$$

Where $f_1, f_2, g_1, g_2, h_1, h_2$ is the constant to be solved, we get:

$$V_I'(G) = \frac{dV_I(G)}{dG} = f_1, \ V_F'(G) = \frac{dV_F(G)}{dG} = g_1, \ V_R'(G) = \frac{dV_R(G)}{dG} = h_1 \tag{39}$$

Substitute the Eqs (38) and (39) into the Eqs (35)–(37), we get:

$$
\begin{aligned}
\rho(f_1 G + f_2) = {}& (\alpha\eta - \delta f_1)G + \frac{(\alpha\varepsilon_R + f_1\lambda_R)((1-\alpha-\beta)\varepsilon_R + h_1\lambda_R)}{\mu_R} \\
& + \frac{k_F\varepsilon_F(2-2\alpha-\beta)(\lambda_F f_1 + \alpha\varepsilon_F) + \lambda_F k_F(g_1 + 2h_1)(f_1\lambda_F + \alpha\varepsilon_F)}{2(2+k_F\mu_F)} \\
& + \frac{k_I(\alpha\varepsilon_I + f_1\lambda_I)((2-\alpha-2\beta)\varepsilon_I + (f_1 + 2h_1)\lambda_I)}{8}
\end{aligned}
\tag{40}
$$

$$
\begin{aligned}
\rho(g_1 G + g_2) = {}& (\beta\eta - \delta g_1)G + \frac{(\beta\varepsilon_R + g_1\lambda_R)((1-\alpha-\beta)\varepsilon_R + h_1\lambda_R)}{\mu_R} \\
& + \frac{g_1 k_F\lambda_F(g_1\lambda_F - 2\varepsilon_F(\alpha-1) + 2h_1\lambda_F) + \beta k_F\varepsilon_F(\varepsilon_F(2-2\alpha-\beta) + 2h_1\lambda_F)}{4(2+k_F\mu_F)} \\
& + \frac{k_I(\beta\varepsilon_I + g_1\lambda_I)((2-\alpha-2\beta)\varepsilon_I + (f_1 + 2h_1)\lambda_I)}{4}
\end{aligned}
\tag{41}
$$

$$
\begin{aligned}
\rho(h_1 G + h_2) = {}& (1-\alpha-\beta)\eta G - \frac{\delta h_1 G}{4} + \frac{2\eta G}{2+k_F\mu_F} + \frac{((1-\alpha-\beta)\varepsilon_R + h_1\lambda_R)^2}{2\mu_R} \\
& + \frac{2k_F(\varepsilon_F(2-2\alpha-\beta) + g_1\lambda_F)(4\lambda_F h_1 + (\varepsilon_F(2-2\alpha-\beta) + g_1\lambda_F)) + 8k_F\lambda_F^2 h_1^2}{16(2+k_F\mu_F)} \\
& + \frac{\varepsilon_I h_1 k_I\lambda_I(2-\alpha-2\beta) + h_1 k_I\lambda_I f_1\lambda_I + h_1^2 k_I\lambda_I^2}{4} + \frac{k_I((2-\alpha-2\beta)\varepsilon_I + f_1\lambda_I)^2}{16}
\end{aligned}
\tag{42}
$$

From the previous assumption that $V_I(G)$, $V_F(G)$ and $V_R(G)$ are satisfied for all $G \geq 0$, we obtain the values of $f_1, f_2, g_1, g_2, h_1, h_2$ as follows:

$$
f_1 = \frac{\alpha\eta}{\rho + \delta}, \; g_1 = \frac{\beta\eta}{\rho + \delta}, \; h_1 = \frac{(1-\alpha-\beta)\eta}{\rho + \delta}
\tag{43}
$$

$$
\begin{aligned}
f_2 = {}& \frac{k_I(\alpha\varepsilon_I + f_1\lambda_I)((2-\alpha-2\beta)\varepsilon_I + (f_1 + 2h_1)\lambda_I)}{8\rho} + \frac{8(\alpha\varepsilon_R + f_1\lambda_R)((1-\alpha-\beta)\varepsilon_R + h_1\lambda_R)}{8\rho\mu_R} \\
& + \frac{4k_F(\alpha\varepsilon_F + f_1\lambda_F)((2-2\alpha-\beta)\varepsilon_F + (g_1 + 2h_1)\lambda_F)}{8\rho(2+k_F\mu_F)}
\end{aligned}
\tag{44}
$$

$$
\begin{aligned}
g_2 = {}& \frac{k_I(\beta\varepsilon_I + g_1\lambda_I)((2-\alpha-2\beta)\varepsilon_I + (f_1 + 2h_1)\lambda_I)}{4\rho} + \frac{(\beta\varepsilon_R + g_1\lambda_R)((1-\alpha-\beta)\varepsilon_R + h_1\lambda_R)}{\rho\mu_R} \\
& + \frac{k_F(\beta\varepsilon_F + g_1\lambda_F)((2-2\alpha-\beta)\varepsilon_F + (g_1 + 2h_1)\lambda_F)}{\rho(8+4k_F\mu_F)}
\end{aligned}
\tag{45}
$$

$$
\begin{aligned}
h_2 = {}& \frac{2k_I((-2+\alpha+2\beta)\varepsilon_I - (f_1 + 2h_1)\lambda_I)^2}{16\rho(2+k_F\mu_F)} + \frac{8((1-\alpha-\beta)\varepsilon_R + h_1\lambda_R)^2}{16\rho\mu_R} \\
& + \frac{k_F(2((2-2\alpha-\beta)\varepsilon_F + (g_1 + 2h_1)\lambda_F)^2 + k_I((2-\alpha-2\beta)\varepsilon_I + (f_1 + 2h_1)\lambda_I)^2\mu_F)}{16\rho(2+k_F\mu_F)}
\end{aligned}
\tag{46}
$$

Substitute $f_1, g_1$, and $h_1$ into the Eqs (30) and (32)–(34) yields the optimal effort levels of innovators, green financial institutions and regulators, and the optimal incentive coefficients

of regulators for innovators and green financial institutions, respectively:

$$E_{I2}^* = \frac{(2 - \alpha - 2\beta)k_I((\delta + \rho)\varepsilon_I + \eta\lambda_I)}{4(\delta + \rho)},$$

$$E_{F2}^* = \frac{(2 - 2\alpha - \beta)k_F((\delta + \rho)\varepsilon_F + \eta\lambda_F)}{2(2 + k_F\mu_F)(\delta + \rho)}$$

$$E_{R2}^* = \frac{(1 - \alpha - \beta)(\varepsilon_R(\delta + \rho) + \eta\lambda_R)}{\mu_R(\delta + \rho)} \tag{47}$$

$$\theta^* = \begin{cases} \dfrac{2 - 3\alpha - 2\beta}{2 - \alpha - 2\beta}, 3\alpha + 2\beta < 2 \\ 0, 3\alpha + 2\beta \geq 2 \end{cases} \tag{48}$$

$$\sigma^* = \begin{cases} \dfrac{2 - 2\alpha - 3\beta}{2 - 2\alpha - \beta}, 2\alpha + 3\beta < 2 \\ 0, 2\alpha + 3\beta \geq 2 \end{cases} \tag{49}$$

Where, since $0 < \theta \leq 1$, $0\ \sigma \leq 1$ and $0 < \alpha\ 1$, $0 < \beta < 1$, can be obtained as $0 < \alpha < \frac{2}{3}$, $0 < \beta < \frac{2}{3}$.

Substitute $f_1, f_2, g_1, g_2, h_1, h_2$ into the Eq (38) yields the optimal payoff functions for the innovator, the green financial institution and the regulator, respectively:

$$V_{I2}(G)^* = \frac{\alpha\eta}{\delta + \rho}G + \frac{1}{8\rho(\delta + \rho)^2}(\alpha k_I(2 - \alpha - 2\beta)((\delta + \rho)\varepsilon_I + \eta\lambda_I)^2$$

$$+ \frac{4\alpha k_F(2 - 2\alpha - \beta)((\delta + \rho)\varepsilon_F + \eta\lambda_F)^2}{(2 + k_F\mu_F)} + \frac{8\alpha(1 - \alpha - \beta)((\delta + \rho)\varepsilon_R + \eta\lambda_R)^2}{\mu_R}) \tag{50}$$

$$V_{F2}(G)^* = \frac{\beta\eta}{(\delta + \rho)}G + \frac{1}{4\rho(\delta + \rho)^2}(\beta k_I(2 - \alpha - 2\beta)((\delta + \rho)\varepsilon_I + \eta\lambda_I)^2$$

$$+ \frac{\beta k_F(2 - 2\alpha - \beta)((\delta + \rho)\varepsilon_F + \eta\lambda_F)^2}{(2 + k_F\mu_F)} + \frac{4\beta(1 - \alpha - \beta)((\delta + \rho)\varepsilon_R + \eta\lambda_R)^2}{\mu_R}) \tag{51}$$

$$V_{R2}(G)^* = \frac{(1 - \alpha - \beta)\eta}{(\delta + \rho)}G + \frac{1}{16\rho(\delta + \rho)^2}(k_I(2 - \alpha - 2\beta)^2((\delta + \rho)\varepsilon_I + \eta\lambda_I)^2$$

$$\frac{2k_F(2 - 2\alpha - \beta)^2((\delta + \rho)\varepsilon_F + \eta\lambda_F)^2}{(2 + k_F\mu_F)} + \frac{8(1 - \alpha - \beta)^2((\delta + \rho)\varepsilon_R + \eta\lambda_R)^2}{\mu_R}) \tag{52}$$

The benefits of the green financial ecosystem are:

$$V(G)^* = \frac{\eta}{(\delta + \rho)}G + \frac{1}{16\rho(\delta + \rho)^2}(k_I(2 - \alpha - 2\beta)(2 + \alpha + 2\beta)((\delta + \rho)\varepsilon_I + \eta\lambda_I)^2$$

$$+ \frac{2k_F(2 - 2\alpha - \beta)(2 + 2\alpha + \beta)((\delta + \rho)\varepsilon_F + \eta\lambda_F)^2}{2 + k_F\mu_F} + \frac{8(1 - \alpha - \beta)(1 + \alpha + \beta)((\delta + \rho)\varepsilon_R + \eta\lambda_R)^2}{\mu_R}) \tag{53}$$

## Synergistic contract

In this case, innovators, green financial institutions and regulators work together to determine the optimal level of effort and optimal benefit function to maximize the benefits of the green financial ecosystem. The effort costs borne by the regulator on behalf of the innovator and the green financial institution are internal fund transfers, and the incentive coefficients $\theta$, $\sigma$ can be any value within the range [0, 1]. At this point, the objective function of the green financial ecosystem is:

$$\max_{E_I(t),E_F(t),E_R(t)} J = \int_0^\infty e^{-\rho t}[\pi(t) - \frac{1}{k_I}E_I^2(t) - (\frac{\mu_F}{2} + \frac{1}{k_F})E_F^2(t) - \frac{\mu_R}{2}E_R^2(t)]dt \tag{54}$$

Suppose that the green financial ecosystem has an optimal return function $V(G)$, which is continuously bounded and differentiable and satisfies the following *HJB* equation for all $G \geq 0$:

$$\rho V(G) = \max_{E_I \geq 0; E_F \geq 0; E_R \geq 0}[\pi(t) - (\frac{\mu_F}{2} + \frac{1}{k_F})E_F^2(t) - \frac{\mu_R}{2}E_R^2(t) + V'(G)(\lambda_I E_I + \lambda_F E_F + \lambda_R E_R - \delta G)] \tag{55}$$

To obtain the right part of the *HJB* equation and maximize it, the condition is to find the first partial derivative of Eq (55) for $E_I$, $E_F$ and $E_R$ respectively, and make it zero. The solution is:

$$E_I = \frac{k_I(\varepsilon_I + \lambda_I V'(G))}{2}, \; E_F = \frac{k_F(\varepsilon_F + \lambda_F V'(G))}{2 + k_F \mu_F}, \; E_R = \frac{\varepsilon_R + \lambda_R V'(G)}{\mu_R} \tag{56}$$

Substitute the Eq (56) into Eq (55) to simplify:

$$\rho V(G) = (\eta - \delta V'(G))G + \frac{k_I(\varepsilon_I + \lambda_I V'(G))^2}{4} + \frac{k_F(\varepsilon_F + \lambda_F V'(G))^2}{2(2 + k_F \mu_F)} + \frac{(\varepsilon_R + V_R \lambda_R)^2}{2\mu_R} \tag{57}$$

From the structure of Eq (57), it can be seen that the unary one-time function $G$ as the independent variable is the solution of the *HJB* equation, so:

$$V(G) = f_1 G + f_2 \tag{58}$$

Where, $f_1, f_2$ is the constant to be solved, it is obtained:

$$V'(G) = \frac{dV(G)}{dG} = f_1 \tag{59}$$

Substitute the Eqs (58) and (59) into the Eq (57), we get:

$$\rho(f_1 G + f_2) = (\eta - \delta f_1)G + \frac{k_I(\varepsilon_I + f_1 \lambda_I)^2}{4} + \frac{k_F(\varepsilon_F + f_1 \lambda_F)^2}{2(2 + k_F \mu_F)} + \frac{(\varepsilon_R + f_1 \lambda_R)^2}{2\mu_R} \tag{60}$$

From the previous assumption, we can see that $V(G)$ is satisfied for all $G \geq 0$, and thus the value of $f_1, f_2$ is:

$$f_1 = \frac{\eta}{\rho + \delta} \tag{61}$$

$$f_2 = \frac{k_I(\varepsilon_I(\delta + \rho) + \eta\lambda_I)^2}{4\rho(\delta + \rho)^2} + \frac{k_F(\varepsilon_F(\delta + \rho) + \eta\lambda_F)^2}{2\rho(\delta + \rho)^2(2 + k_F \mu_F)} + \frac{(\varepsilon_R(\delta + \rho) + \eta\lambda_R)^2}{2\rho\mu_R(\delta + \rho)^2} \tag{62}$$

Substitute $f_1$ into Eq (56), the optimal effort level of the innovator, green financial institution and regulatory party is respectively:

$$E_{I3}^* = \frac{k_I(\varepsilon_I(\delta+\rho)+\eta\lambda_I)}{2(\delta+\rho)}, \ E_{F3}^* = \frac{k_F(\varepsilon_F(\delta+\rho)+\eta\lambda_F)}{(2+k_F\mu_F)(\delta+\rho)}, \ E_{R3}^* = \frac{\varepsilon_R(\delta+\rho)+\eta\lambda_R}{\mu_R(\delta+\rho)} \tag{63}$$

Substitute $f_1, f_2$ into Eq (58), the optimal income function of the green financial ecosystem can be obtained as follows:

$$V_3(G)^* = \frac{\eta}{(\delta+\rho)}G + \frac{k_I(\varepsilon_I(\delta+\rho)+\eta\lambda_I)^2}{4\rho(\delta+\rho)^2} + \frac{k_F(\varepsilon_F(\delta+\rho)+\eta\lambda_F)^2}{2\rho(2+k_F\mu_F)(\delta+\rho)^2} + \frac{(\varepsilon_R(\delta+\rho)+\eta\lambda_R)^2}{2\rho\mu_R(\delta+\rho)^2} \tag{64}$$

The optimal income function of the innovator, green financial institution and regulator is as follows:

$$V_{I3}(G)^* = \frac{\alpha\eta}{(\delta+\rho)}G + \frac{\alpha k_I(\varepsilon_I(\delta+\rho)+\eta\lambda_I)^2}{4\rho(\delta+\rho)^2} + \frac{\alpha k_F(\varepsilon_F(\delta+\rho)+\eta\lambda_F)^2}{2\rho(2+k_F\mu_F)(\delta+\rho)^2} + \frac{\alpha(\varepsilon_R(\delta+\rho)+\eta\lambda_R)^2}{2\rho\mu_R(\delta+\rho)^2} \tag{65}$$

$$V_{F3}(G)^* = \frac{\beta\eta}{(\delta+\rho)}G + \frac{\beta k_I(\varepsilon_I(\delta+\rho)+\eta\lambda_I)^2}{4\rho(\delta+\rho)^2} + \frac{\beta k_F(\varepsilon_F(\delta+\rho)+\eta\lambda_F)^2}{2\rho(2+k_F\mu_F)(\delta+\rho)^2} + \frac{\beta(\varepsilon_R(\delta+\rho)+\eta\lambda_R)^2}{2\rho\mu_R(\delta+\rho)^2} \tag{66}$$

$$V_{R3}(G)^* = \frac{(1-\alpha-\beta)\eta}{(\delta+\rho)}G + \frac{(1-\alpha-\beta)k_I(\varepsilon_I(\delta+\rho)+\eta\lambda_I)^2}{4\rho(\delta+\rho)^2} + \frac{(1-\alpha-\beta)k_F(\varepsilon_F(\delta+\rho)+\eta\lambda_F)^2}{2\rho(2+k_F\mu_F)(\delta+\rho)^2}$$
$$+ \frac{(1-\alpha-\beta)(\varepsilon_R(\delta+\rho)+\eta\lambda_R)^2}{2\rho\mu_R(\delta+\rho)^2} \tag{67}$$

## Comparative analysis of three contract

This paper obtains the following conclusions by comparing and analyzing the optimal level of effort, the most benefit function and the optimal benefit function of the green financial ecosystem of innovators, green financial institutions and regulators in the three contracts.

**Proposition 1:** Compared to the No-incentive Contract, the effort level of the regulator under the Cost-sharing Contract remains unchanged, and the effort level of both the innovator and the green financial institution increases by an amount equal to the proportion of the regulator's subsidy of its costs, i.e., the optimal incentive coefficient. The three types of subjects have the highest level of effort under the Synergistic Contract, i.e., $E_{I1}^* < E_{I2}^* < E_{I3}^*$, $E_{F1}^* < E_{F2}^* < E_{F3}^*$, $E_{R1}^* = E_{R2}^* < E_{R3}^*$, $\frac{E_{I2}^*-E_{I1}^*}{E_{I2}^*} = \theta^*$, $\frac{E_{F2}^*-E_{F1}^*}{E_{F2}^*} = \sigma^*$.

**Proof:**

From Eqs (23), (47)–(49) and (63), it can be obtained (where, $0 < \alpha < \frac{2}{3}, 0 < \beta < \frac{2}{3}$):

$$
\begin{aligned}
E_{I2}^* - E_{I1}^* &= \frac{(2 - 3\alpha - 2\beta)k_I((\delta + \rho)\varepsilon_I + \eta\lambda_I)}{4(\delta + \rho)} \\
&= \frac{(2 - \alpha - 2\beta)k_I((\delta + \rho)\varepsilon_I + \eta\lambda_I)}{4(\delta + \rho)} \cdot \frac{2 - 3\alpha - 2\beta}{2 - \alpha - 2\beta} = E_{I2}^* \cdot \theta^* > 0
\end{aligned}
\tag{68}
$$

$$
\begin{aligned}
E_{F2}^* - E_{F1}^* &= \frac{(2 - 2\alpha - 3\beta)k_F((\delta + \rho)\varepsilon_F + \eta\lambda_F)}{2(\delta + \rho)(2 + k_F\mu_F)} \\
&= \frac{(2 - 2\alpha - \beta)k_F((\delta + \rho)\varepsilon_F + \eta\lambda_F)}{2(2 + k_F\mu_F)(\delta + \rho)} \cdot \frac{2 - 2\alpha - 3\beta}{2 - 2\alpha - \beta} = E_{F2}^* \cdot \sigma^* > 0
\end{aligned}
\tag{69}
$$

$$
E_{R1}^* = E_{R2}^* = \frac{(1 - \alpha - \beta)(\varepsilon_R(\delta + \rho) + \eta\lambda_R)}{\mu_R(\delta + \rho)}
\tag{70}
$$

$$
E_{I3}^* - E_{I2}^* = \frac{(\alpha + 2\beta)k_I((\delta + \rho)\varepsilon_I + \eta\lambda_I)}{4(\delta + \rho)} > 0
\tag{71}
$$

$$
E_{F3}^* - E_{F2}^* = \frac{(2\alpha + \beta)k_F((\delta + \rho)\varepsilon_F + \eta\lambda_F)}{2(\delta + \rho)(2 + k_F\mu_F)} > 0
\tag{72}
$$

$$
E_{R3}^* - E_{R2}^* = \frac{(\alpha + \beta)((\delta + \rho)\varepsilon_R + \eta\lambda_R)}{(\delta + \rho)\mu_R} > 0
\tag{73}
$$

**End of proof**.

**Proposition 2**: The optimal benefit function for all three types of subjects under the Cost-sharing Contract is greater than the Non-incentive Contract, i.e., $V_{I2}(G)^* \geq V_{I2}(G)^*$, $V_{F2}(G)^* \geq V_{F1}(G)^*$, $V_{R2}(G)^* \geq V_{R1}(G)^*$.

**Proof**:

From Eqs (24)–(26) and (48)–(52), it can be obtained (where, $0 < \alpha < \frac{2}{3}, 0 < \beta < \frac{2}{3}$):

$$
\begin{aligned}
V_{I2}(G)^* - V_{I1}(G)^* &= \frac{1}{8\rho(\delta + \rho)^2}(\alpha k_I(2 - 3\alpha - 2\beta)((\delta + \rho)\varepsilon_I + \eta\lambda_I)^2 \\
&+ \frac{4\alpha k_F(2 - 2\alpha - 3\beta)((\delta + \rho)\varepsilon_F + \eta\lambda_F)^2}{(2 + k_F\mu_F)}) > 0
\end{aligned}
\tag{74}
$$

$$
\begin{aligned}
V_{F2}(G)^* - V_{F1}(G)^* &= \frac{1}{4\rho(\delta + \rho)^2}(\beta k_I(2 - 3\alpha - 2\beta)((\delta + \rho)\varepsilon_I + \eta\lambda_I)^2 \\
&+ \frac{\beta k_F(2 - 2\alpha - 3\beta)((\delta + \rho)\varepsilon_F + \eta\lambda_F)^2}{(2 + k_F\mu_F)}) > 0
\end{aligned}
\tag{75}
$$

$$
\begin{aligned}
V_{R2}(G)^* - V_{R1}(G)^* &= \frac{1}{16\rho(\delta + \rho)^2}(k_I(2 - 3\alpha - 2\beta)^2((\delta + \rho)\varepsilon_I + \eta\lambda_I)^2 \\
&+ \frac{2k_F(2 - 2\alpha - 3\beta)^2((\delta + \rho)^2\varepsilon_F + \eta\lambda)^2}{(2 + k_F\mu_F)}) > 0
\end{aligned}
\tag{76}
$$

**End of proof**.

**Proposition 3**: The optimal benefits of green finance ecosystems are maximized under the Synergistic Contract, followed by the Cost-sharing Contract, and minimized under the Non-incentive Contract, i.e., $V_1(G)^* < V_2(G)^* < V_3(G)^*$.

**Proof**:

From Eqs (27), (48), (49), (53) and (64), it can be obtained (where, $0 < \alpha < \frac{2}{3}, 0 < \beta < \frac{2}{3}$):

$$V_2(G)^* - V_1(G)^* = \frac{1}{16\rho(\delta + \rho)^2}((2 - 3\alpha - 2\beta)(2 - \alpha + 2\beta))k_I((\delta + \rho)\varepsilon_I + \eta\lambda_I)^2$$

$$+ \frac{2k_F}{(2 + k_F\mu_F)}(2 - 2\alpha - 3\beta)(2 + 2\alpha - \beta)((\delta + \rho)\varepsilon_F + \eta\lambda_F)^2) > 0 \tag{77}$$

$$V_3(G)^* - V_2(G)^* = \frac{1}{16\rho(\delta + \rho)^2}(k_I(\alpha + 2\beta)^2((\delta + \rho)\varepsilon_I + \eta\lambda_I)^2$$

$$+ \frac{2k_F(2\alpha + \beta)^2((\delta + \rho)\varepsilon_F + \eta\lambda_F)^2}{(2 + k_F\mu_F)} + \frac{8(\alpha + \beta)^2((\delta + \rho)\varepsilon_R + \eta\lambda_R)^2}{\mu_R}) > 0 \tag{78}$$

**End of proof**.

The optimal benefits for the innovator, the green financial institution and the regulator are all enhanced in the Cost-sharing Contract compared to the Non-incentive Contract, which is Pareto Efficient. The overall benefits to the green finance ecosystem are optimal under the Synergistic Contract, next under the Cost-sharing Contract, and lowest under the Non-incentive Contract. Therefore, the Synergistic Contract aiming at maximizing the overall benefits of the green financial ecosystem is the optimal solution to improve the level of green financial risk management, which can effectively enhance the interests of all parties and the whole and achieve Pareto Optimality.

## Model simulation

The outcomes of innovators, green financial institutions and regulators in terms of their respective levels of green input efforts, returns, and overall returns to the green financial ecosystem under the three contracts depend on the parameter assignments. Combined with the parameter assignment logic of scholars [30], and based on the constraints of the model, the range of parameter values and the actual situation of the green financial market, the parameter values are given, and the results of the model are compared by way of arithmetic example analysis to verify the research conclusions. The parameter assignments in this paper are shown in Table 1.

According to the model formula, the results of the optimal green input effort level, benefit level, and overall benefit level of the green financial ecosystem of the innovator, green financial institution, and regulator under the three contracts of Non-incentive, Cost-sharing, and Synergy are calculated, as shown in Table 2.

**Table 1. Parameters and values.**

| Parameter | Value | Parameter | Value | Parameter | Value | Parameter | Value |
|---|---|---|---|---|---|---|---|
| $G(0)$ | 5 | $\lambda_F$ | 0.4 | $\varepsilon_R$ | 0.7 | $\rho$ | 0.1 |
| $\mu_F$ | 0.6 | $\lambda_R$ | 0.3 | $k_I$ | 0.4 | $\delta$ | 0.1 |
| $\mu_R$ | 0.4 | $\varepsilon_I$ | 0.8 | $k_F$ | 0.2 | $\alpha$ | 0.3 |
| $\lambda_I$ | 0.4 | $\varepsilon_F$ | 0.9 | $\eta$ | 0.4 | $\beta$ | 0.2 |

**Table 2. Comparative analysis of equilibrium results under three contracts.**

|  | Non-incentive Contract | Cost-sharing Contract | Synergy Contract |
|---|---|---|---|
| $E_I$ | 0.096 | 0.202 | 0.32 |
| $E_F$ | 0.0320755 | 0.0962264 | 0.160377 |
| $E_R$ | 1.625 | 1.625 | 3.25 |
| $G$ | $5.3873 - 0.3873e^{-0.1t}$ | $6.06791 - 1.0679e^{-0.1t}$ | $11.6715 - 6.6715e^{-0.1t}$ |
| $V_I$ | $0.6G + 6.73148$ | $0.6G + 7.32745$ | $0.6G + 7.51446$ |
| $V_F$ | $0.4G + 2.19775$ | $0.4G + 5.05418$ | $0.4G + 5.00964$ |
| $V_R$ | $G + 6.32189$ | $G + 6.8536$ | $G + 12.5241$ |
| $V$ | $2G + 15.25112$ | $2G + 19.23523$ | $2G + 25.0482$ |

## Comparison of optimal returns of three contracts

The optimal benefit function is shown in Figs 3–6, where the benefits to the innovator, the green financial institution and the regulator increase with time under the Cost-sharing Contract and the Synergistic Contract. Synergistic Contracts are superior to Cost-sharing Contracts, both for individual participants and for the green finance ecosystem as a whole, and both are superior to Non-incentive contracts. The results of this simulation are consistent with the conclusions of Proposition 2 and Proposition 3.

## Impact of important variables in the synergistic contract

**Impact of the innovative capacity factor $\lambda_i(i = I, F, R)$.** According to the model formula, the results of the impact of the level of effort m invested by the three subjects on the level of

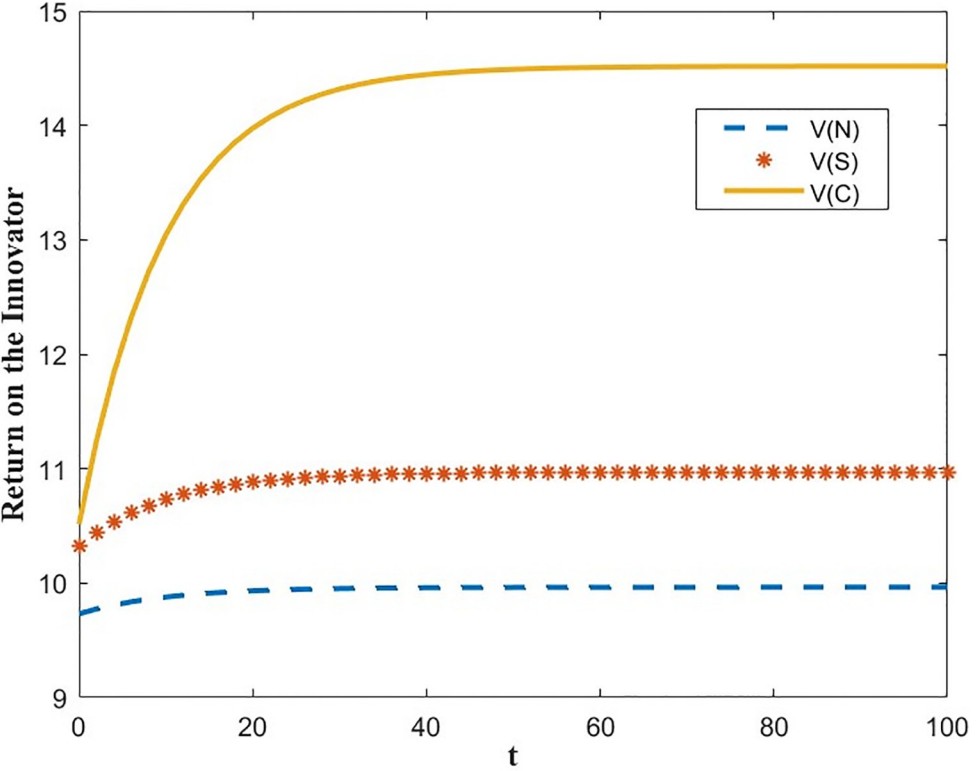

**Fig 3. Optimal return of the innovator.**

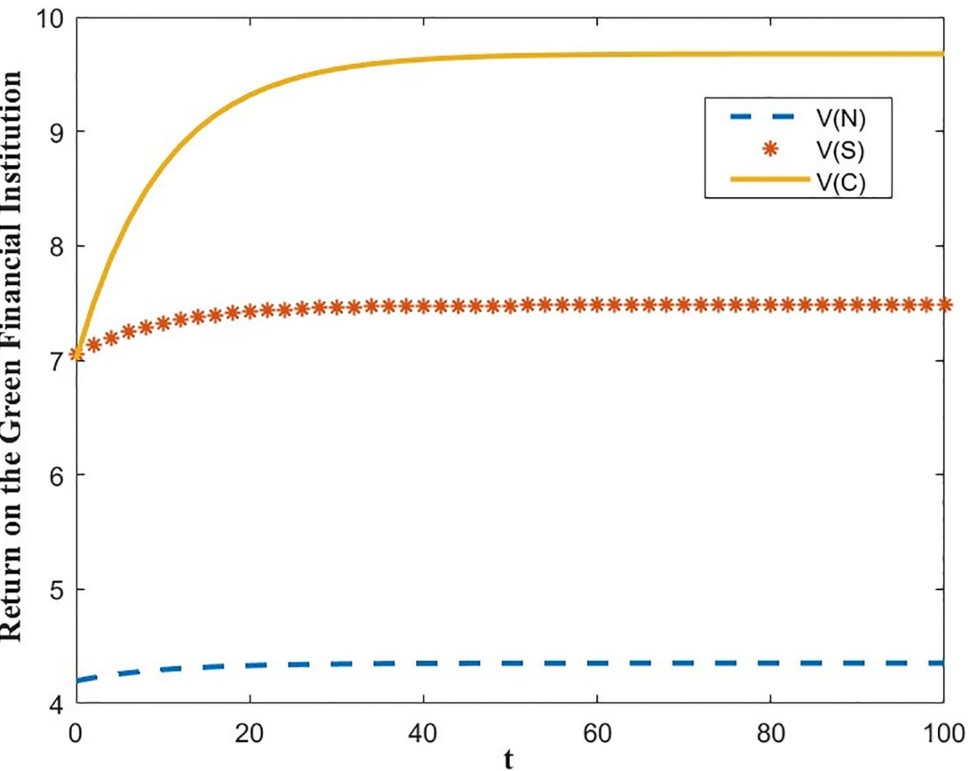

**Fig 4. Optimal returns of the green financial institution.**

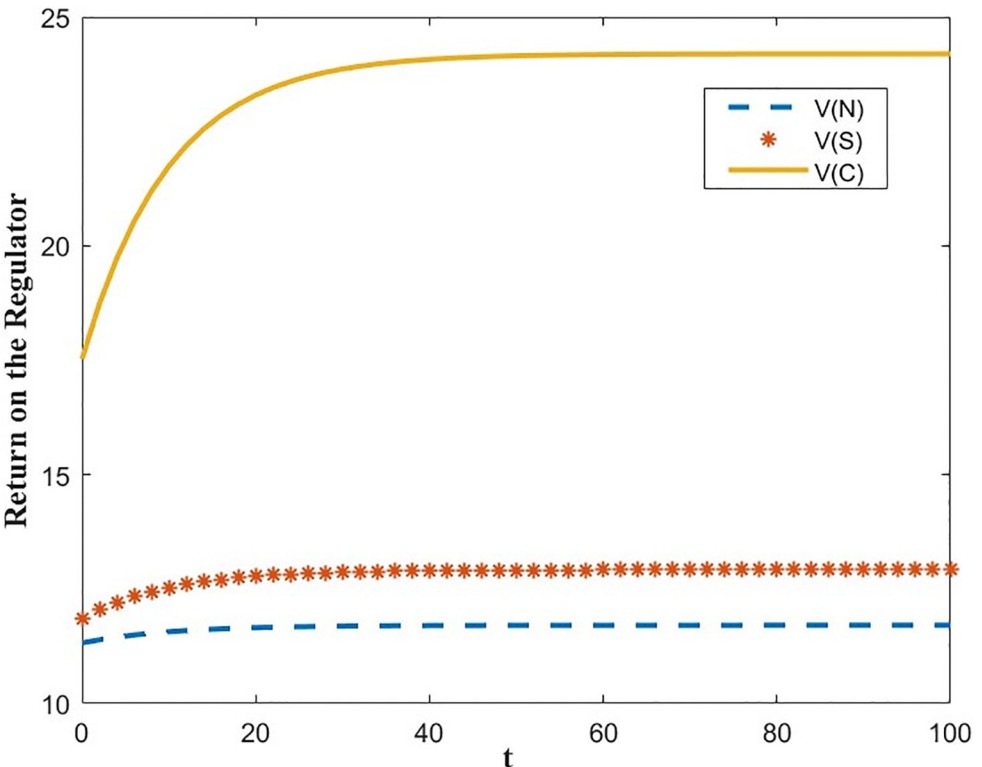

**Fig 5. Optimal returns of the regulators.**

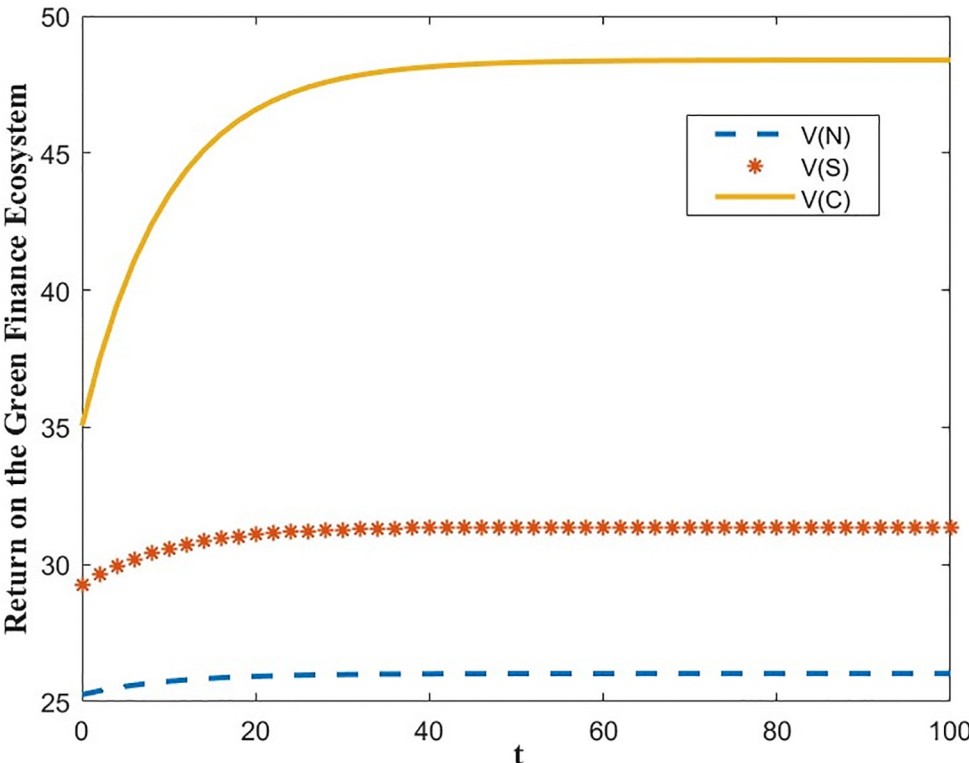

**Fig 6. Optimal returns of the green financial ecosystems.**

green financial ecosystem benefits under the Synergistic Contract are calculated and shown in Table 3.

From Eq (64), $\frac{\partial V_3(k)^*}{\partial \lambda_i} > 0$, i.e., the level of effort $\lambda_i$ invested by the three subjects has a positive effect on the green financial ecosystem return $V_3(k)^*$. The effect of $\lambda_i$ and $t$ on $V_3(k)^*$ is shown in Figs 7 and 8. The level of effort invested by innovators and green financial institutions has a negligible effect on returns in the early part of the period, with a gradually increasing effect in the later. The effort invested by the regulator has the most significant impact on returns.

**Impact of the human resources cost factor $\mu_i(i = F, R)$.** According to the model formula, the results of the impact of the human resource cost factor m invested by green financial institutions and regulators on the level of green financial ecosystem benefits under the Synergistic Contract are calculated and shown in Table 4.

From Eq (64), $\frac{\partial V_3(k)^*}{\partial \mu_i} < 0$, i.e., the human resource cost coefficients $\mu_i$ of green financial institutions and regulators hurt the green financial ecosystem return $V_3(k)^*$. The effect of $\mu_i$ and $t$ on $V_3(k)^*$ is shown in Figs 9 and 10. Green financial institutions have a more negligible impact on returns, the regulator's human resource costs have the most significant impact on

**Table 3. Results of the effect of $\lambda_i$ on the level of green financial ecosystem return.**

| | |
|---|---|
| $V_{\lambda_I}$ | $45.8312 + 25(0.16 + 0.4\lambda_I)^2 - 13.343e^{-0.1t}$ |
| $V_{\lambda_F}$ | $47.028 + 11.7925(0.18 + 0.4\lambda_F)^2 - 13.343e^{-0.1t}$ |
| $V_{\lambda_R}$ | $27.26621 + 312.5(0.14 + 0.4\lambda_R)^2 - 13.343e^{-0.1t}$ |

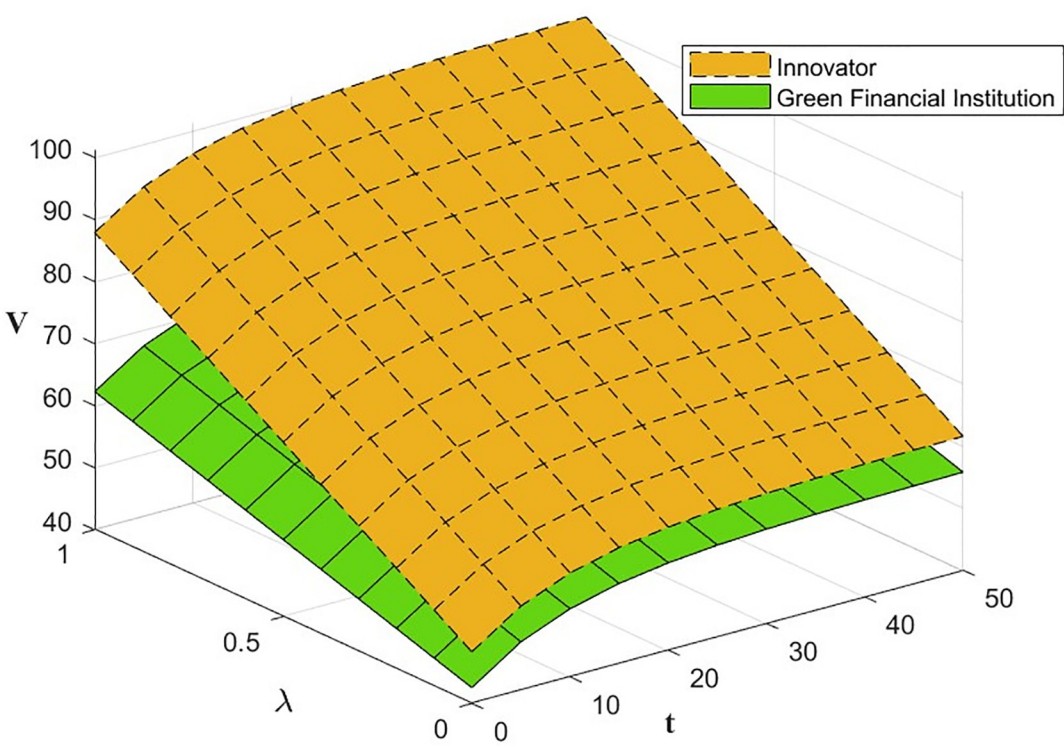

**Fig 7. The effect of $\lambda_I$ and $\lambda_F$ on $V_3(k)^*$.**

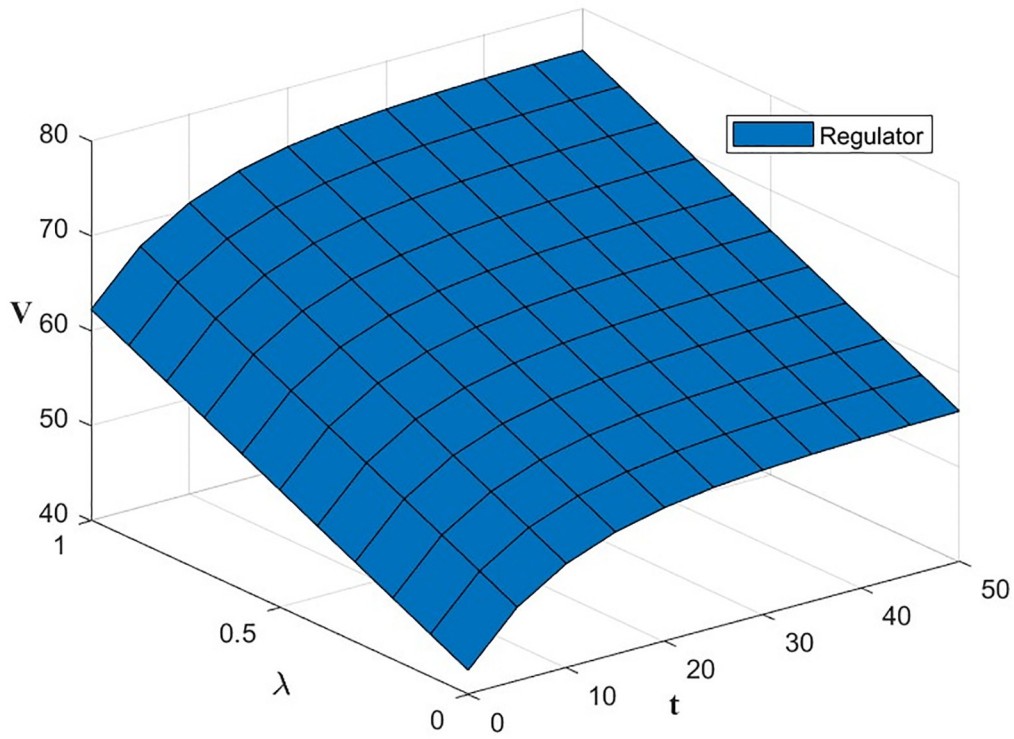

**Fig 8. The effect of $\lambda_R$ on $V_3(k)^*$.**

**Table 4. Results of the effect of $\mu_i$ on the level of green financial ecosystem return.**

| | |
|---|---|
| $V_{\mu_F}$ | $47.028 - 13.343\mathrm{e}^{-0.1t} + \frac{2.89}{(2+0.2\mu_F)}$ |
| $V_{\mu_R}$ | $27.26621 - 13.343\mathrm{e}^{-0.1t} + \frac{8.45}{\mu_R}$ |

returns, and their impact deepens over time, with a more negligible $\mu_R$ in the early stages of the partnership, which has a more negligible impact on returns. Later in the partnership, as $\mu_R$ increases, returns decrease more rapidly.

**Impact of the technology innovation cost factor $k_i (i = I, F)$.** According to the model formula, the results of the impact of the technological innovation cost factor $k_i$ of the innovator and the green financial institution on the level of return of the green financial ecosystem under the Synergistic Contract are calculated as shown in Table 5.

From Eq (64), $\frac{\partial V_3(k)^*}{\partial k_i} > 0$, i.e., the technological innovation cost factor $k_i$ of the innovator and the green financial institution has a positive effect on the green financial ecosystem return $V_3(k)^*$. The effect of $k_i$ and $t$ on $V_3(k)^*$ is shown in Figs 11 and 12. As the technological innovation cost factor $k$ increases, $V_3(k)^*$ decreases slightly, but not significantly. The results of this study suggest that although the application of novel technologies in new types of green finance incurs higher technology costs, it only reduces system returns by a small margin. Therefore, the development of risk management in green financial ecosystems should assess the application value of novel technologies from the perspective of user needs [31].

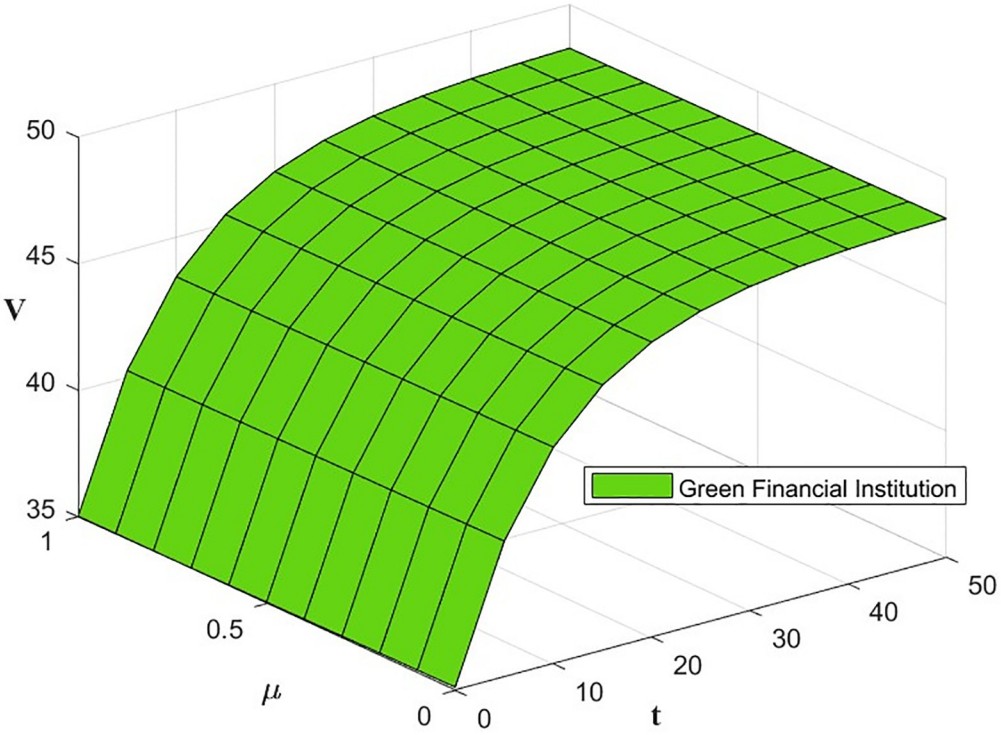

**Fig 9. The effect of $\mu_F$ on $V_3(k)^*$.**

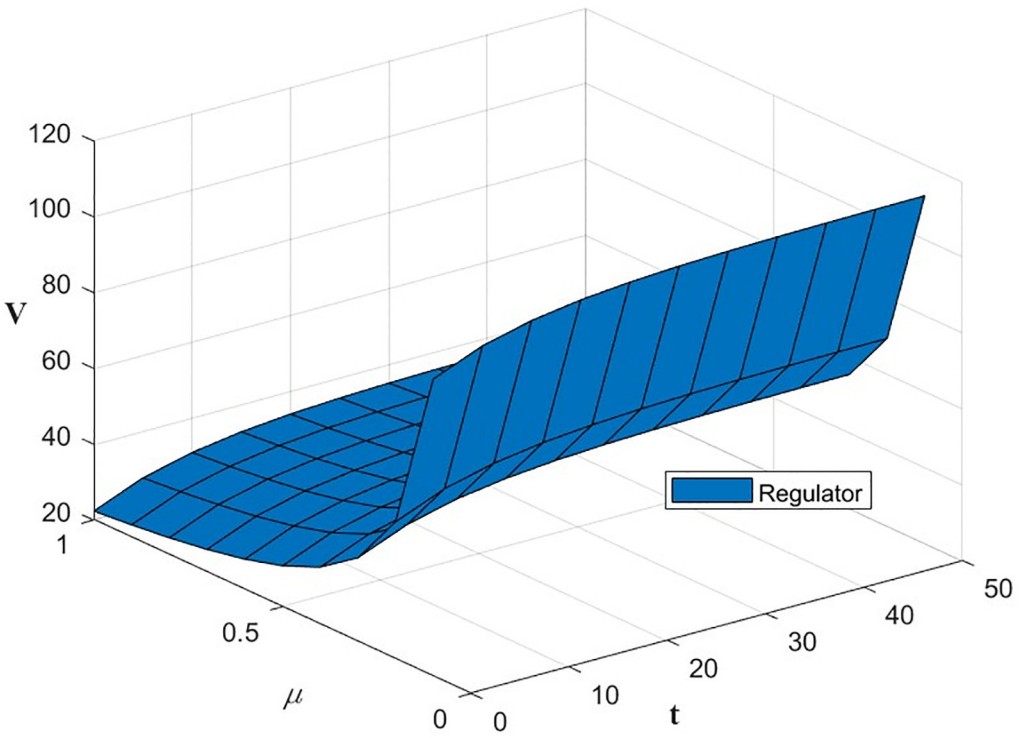

**Fig 10. The effect of $\mu_R$ on $V_3(k)^*$.**

**Table 5. Results of the effect of $k_i$ on the level of green financial ecosystem return.**

| | |
|---|---|
| $V_{k_I}$ | $45.8312 - 13.343\mathrm{e}^{-0.1t} + 6.4k_I$ |
| $V_{k_F}$ | $47.028 - 13.343\mathrm{e}^{-0.1t} + \frac{14.45k_F}{(2+0.6k_F)}$ |

## Conclusions and implications

### Main conclusions

This paper utilizes a differential game model to study the cooperation among innovators, green financial institutions and regulators in the risk management of green financial ecosystems and obtain the optimal level of effort, the optimal return and the optimal return of green financial ecosystems under the Non-incentive Contract, the Cost-sharing Contract and the Synergistic Contract of the three types of subjects respectively.

By comparing the equilibrium solutions and simulation results under the three contracts, the following conclusions are obtained:

1. The incentives under the Cost-sharing Contract and the Synergy Contract are effective, and both raise the level of effort of the subjects to varying degrees. In particular, the level of effort of the three types of subjects is highest under the Synergistic Contract. At the same time, the incentive intensity under the Cost-sharing Contract is equal to the proportion of costs subsidized by the regulator to the innovator and the green financial institution. Therefore, regulators can flexibly adjust their subsidies to innovators and green financial institutions according to their needs and changes in the green finance market environment.

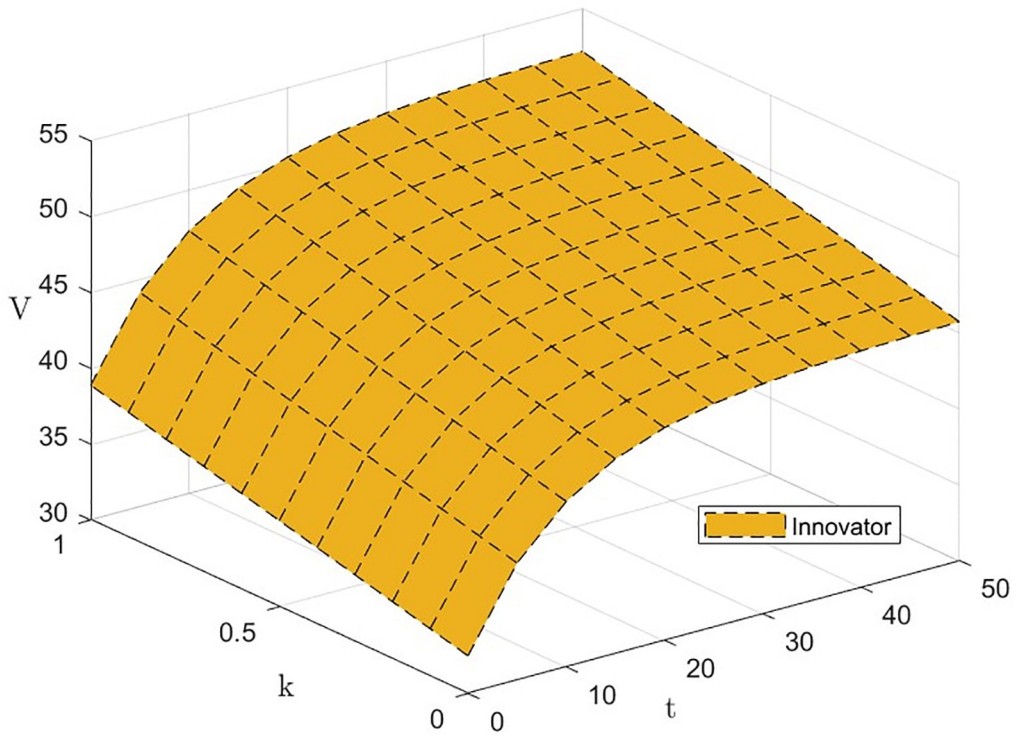

**Fig 11. The effect of $k_I$ on $V_3(k)^*$.**

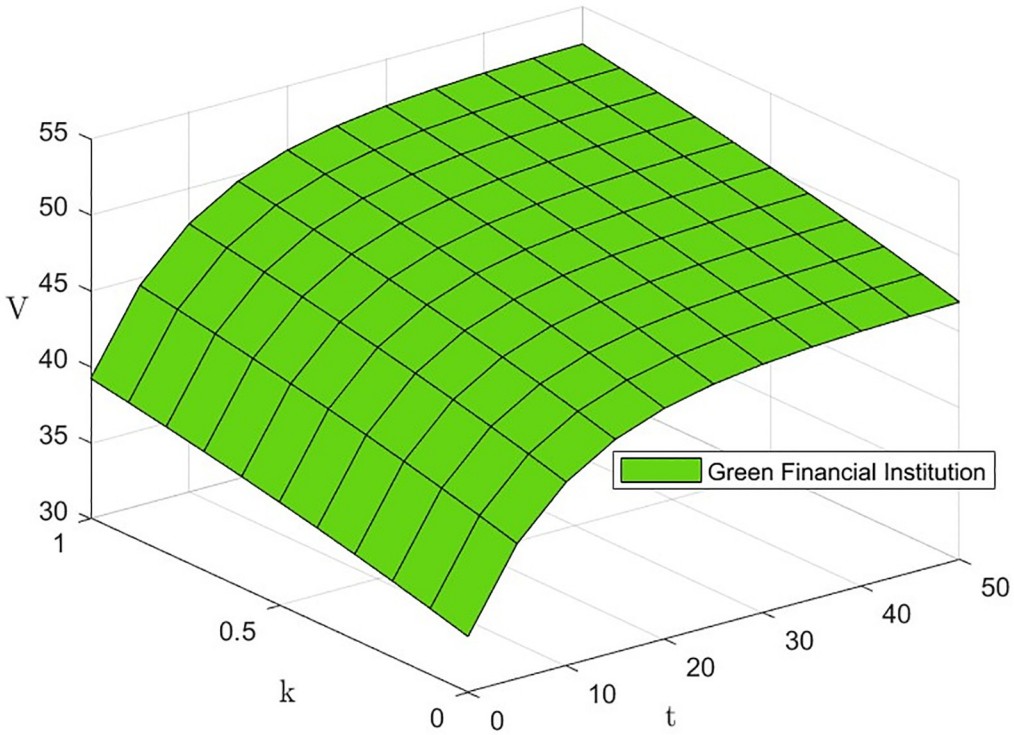

**Fig 12. The effect of $k_F$ on $V_3(k)^*$.**

2. Under the Synergistic Contract, the overall return of the green financial ecosystem increases. In contrast, the return of the three types of subjects improves, which is the development direction of building the risk management of the green financial ecosystem. However, if the return of the three types of subjects is not reasonably divided, the return of the three types of subjects will be lower than that of the Non-incentive contract. Therefore, to achieve the Pareto Optimality of each subject and the whole green financial ecosystem simultaneously, the value of the benefit distribution coefficient should be set reasonably.

3. In order to improve the overall return of the green financial ecosystem, the three types of the subject should take targeted measures, such as improving the level of technological innovation, reducing the labor cost coefficient and increasing marginal returns.

4. The impact of technological innovation costs on the overall benefits of the green financial ecosystem is small. Therefore, although there are many technical challenges in the research and development of products by innovators and green financial institutions, the value potential is still very significant from the ecological value of green product and service innovation and risk management in green financial ecosystems.

### Theoretical implications

The research process of this dissertation has yielded the following theoretical implications for the study of risk management in green financial ecosystems:

1. Applying the differential game approach to green financial ecosystem risk management deepens the explanatory strength of the level of effort and technological innovation of innovators, green financial institutions and regulators in green financial ecosystem research. The differential game approach explores the three contractual models in the evolutionary process of green financial ecosystem risk management, providing a more nuanced explanation of green financial ecosystem risk management. At the same time, the model simulation intuitively analyzes the impact of essential variables on the evolution of risk management in green financial ecosystems, which responds to the scholars' call to promote the sustainable development of green financial ecosystems by adopting a new methodology to examine the evolution process and path of exploratory and exploitative green financial risk management [32].

2. Combining the theory of green financial ecosystem with the theory of green financial risk management deepens the research on green financial risk management from the perspective of complex systems. On the one hand, the core subjects in the risk management of green financial ecosystems are analyzed based on the hierarchical structure, which not only deepens the synergistic mechanism of different subjects in the risk management of green ecosystems but also responds to the scholars' call for the use of the green financial ecosystem perspective to study the evolution of the field of green financial risk management [33]. On the other hand, based on the influence of multiple variables on the participating subjects and green financial ecosystem benefits, the differences in strategy selection of different subjects in the green financial risk management evolution model are further identified, which provides an essential reference for the high-quality development of risk management in green financial ecosystems, responding to the scholars' analysis of the influencing elements in the process of exploratory and exploitative green financial risk management evolution [34].

## Management implications

The findings of this paper draw the following management implications for the development of risk management in green financial ecosystems:

1.  It is utilizing green financial technology innovation to build a green financial ecosystem. The research in this paper shows that the contractual model of collaborative cooperation among innovators, green financial institutions and regulators, which aims at maximizing the overall benefits of the green financial ecosystem, is the optimal solution for promoting the development of green financial risk management. It can effectively enhance the efforts and benefits of all the participating subjects.

2.  The regulator establishes sound green financial policies and regulations, rewards, punishments and incentives. The study concluded that green finance is characterized by relatively significant externalities, and how to internalize such externalities is a crucial issue, which determines that the development of green risk management requires a solid policy-driven drive for better functioning of the green financial ecosystem [35].

3.  Green financial institutions must strengthen their innovation of green financial products centered on user needs. The study concludes that the efforts of green financial institutions to invest and innovate technologically can increase the willingness of regulators to provide them with incentives and improve risk management in the green financial ecosystem.

## Research gaps and future prospects

This paper divides the participants in the green financial ecosystem into three types of subjects. However, the same subject within the inevitable also involves a more complex competitive relationship. In order to simplify the model, this paper does not model analysis of this problem. Therefore, the complex green financial ecosystem risk management of the problem of cooperation needs to be further studied. In addition to the incentives studied in this paper, there are factors like green goodwill [36], green business process maturity [37], and green startup maturity [38] that affect the level of effort of the participants. In addition, with the increase of green finance user data and the number of green finance products, building the optimal structural composition of the green finance risk management system so that the participating subjects can continue to create value in a stable state also needs to be explored in depth. Future research could focus on such issues and try to study them from the perspectives of more complex competing relationships, multiple incentives, etc. In addition, the optimal structure and subject relationship of risk management in the green financial ecosystem can be studied so that the innovators can continue to promote high-quality risk management in a stable state and thus create more value.

## Author Contributions

**Conceptualization:** Shuang Lu, JinRong Liu.

**Data curation:** Shuang Lu.

**Formal analysis:** Shuang Lu.

**Funding acquisition:** ZhongPing Cui.

**Methodology:** ZhongPing Cui.

**Project administration:** ZhongPing Cui.

**Software:** Shuang Lu.

**Supervision:** Shuang Lu.

**Validation:** Shuang Lu, JinRong Liu.

**Visualization:** ZhongPing Cui, Shuang Lu.

**Writing – original draft:** Shuang Lu.

**Writing – review & editing:** Shuang Lu, JinRong Liu.

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
