## [Decision Letter · Decision Letter 0]

3 Jan 2024

PONE-D-23-37731Research on Risk Management Incentive Strategy Based on the Green Financial EcosystemPLOS ONE

Dear Dr. Lu,

Thank you for submitting your manuscript to PLOS ONE. After careful consideration, we feel that it has merit but does not fully meet PLOS ONE’s publication criteria as it currently stands. Therefore, we invite you to submit a revised version of the manuscript that addresses the points raised during the review process.

We look forward to receiving your revised manuscript.

Kind regards,

Xin Shen

Academic Editor

PLOS ONE

Journal Requirements:

Reviewers' comments:

Reviewer's Responses to Questions

**Comments to the Author**

1. Is the manuscript technically sound, and do the data support the conclusions?

Reviewer #1: Yes

Reviewer #2: Yes

2. Has the statistical analysis been performed appropriately and rigorously? 

Reviewer #1: Yes

Reviewer #2: Yes

3. Have the authors made all data underlying the findings in their manuscript fully available?

Reviewer #1: Yes

Reviewer #2: Yes

4. Is the manuscript presented in an intelligible fashion and written in standard English?

Reviewer #1: No

Reviewer #2: Yes

5. Review Comments to the Author

Reviewer #1: This is an interesing research about green financial ecosystem risk management incentive strategy. The method of game model is applicable. The basic research process is complete. However, there are some questions needed to be responsed as follows.

(1) clarify the research gap in the introduction section, do not write it in the last prograph;

(2) add literature review. You can move some contents from the part of introduction but make sure to rewrite this part, because literature review in the original introduction part is just a list of reference with short depicion

(3) I dont believe "Differential Game Model" is a good section title. Please change it.

(4) It seems that Model Assumption does not exactly match research questions.

(5) In Model Simulation, parameter values are set without reason. Do they realize the simulaiton of reality? It looks more like an experiment report.

Reviewer #2: The paper focuses on improving efforts and benefits of risk management in the green financial ecosystem, involving innovators, green financial institutions, and regulators. Three contract modes (No-incentive Contract, Cost-sharing Contract, Synergistic Cooperation Contract) are explored for optimal effort levels and returns on risk management. The findings highlight increased incentives and returns under the latter two contracts, with Synergistic Cooperation Contract showing the highest effort levels and Pareto optimization. The research enhances understanding of cooperation in risk management and provides practical insights for stakeholders in green financial ecosystems.

The paper systematically investigates three contract modes, providing insights into their impact on incentives, effort levels, and returns. This analysis contributes to the understanding of contractual mechanisms in the green financial ecosystem. The use of numerical simulation analysis adds empirical support to the theoretical findings, enhancing the robustness of the results. This approach strengthens the credibility of the proposed optimal levels and their implications. The identification of the Synergistic Cooperation Contract as achieving Pareto optimization is a significant contribution. It implies a win-win situation for all participating subjects and the green financial ecosystem, emphasizing the importance of collaborative efforts. The study's findings hold practical relevance for risk managers in green financial ecosystems, offering realistic references for decision-making. This practical applicability enhances the impact of the research.

However, while the paper acknowledges the importance of green financial risk management, there is a lack of in-depth exploration of specific risks and their mitigation strategies. A more detailed discussion would provide a holistic view of challenges in the green financial ecosystem, and a qualitative discussion of other approaches that have been introduced in the literature, using similar models, such as A Paradigm Shift Toward Satisfaction, Realism and Efficiency in Wireless Networks Resource Sharing, doi: 10.1109/MNET.011.2000368, is needed. The paper introduces three contract modes but does not delve deeply into the theoretical mechanisms underlying each mode. A more thorough theoretical discussion would enhance the reader's understanding of the contractual dynamics. The introduction provides a qualitative overview of previous studies on green finance, but a more structured presentation, perhaps in the form of a literature review, could enhance the coherence and depth of the introduction. The paper lacks details on the reproducibility of the numerical simulation analysis. Providing sufficient information on parameters and methodologies would enable other researchers to validate and build upon the results.

The paper contributes to the field by examining contractual modes in the green financial ecosystem and identifying the Synergistic Cooperation Contract as an optimal approach. The novelty lies in the combination of contractual analysis and numerical simulations. While the numerical simulation adds empirical support, a more detailed explanation of the simulation methodology, parameters, and sensitivity analyses would enhance the technical depth of the paper. This information is crucial for understanding the robustness of the findings. The paper is generally well-structured, but the introduction could benefit from a more organized presentation of previous studies on green finance. Additionally, clearer visual aids or tables summarizing key numerical results would improve the presentation's clarity. The conclusion briefly mentions the contributions but could elaborate on potential avenues for future research. Suggestions for further investigations or extensions of the current model would enhance the paper's completeness.

In summary, the paper makes valuable contributions to understanding contract modes in the green financial ecosystem. Addressing the highlighted points would further strengthen its impact and provide a more comprehensive perspective on green financial risk management.

6. PLOS authors have the option to publish the peer review history of their article (what does this mean?). If published, this will include your full peer review and any attached files.

Reviewer #1: No

Reviewer #2: No

---

## [Author Response · Author response to Decision Letter 0]

12 Jan 2024

To Reviewer 1:

Thank you very much for your precious comments on this paper, which will improve it greatly. We have made serious revisions, and below are the specific revised responses.

Comment 1. Clarify the research gap in the introduction section, do not write it in the last paragraph.

Response: We are very grateful for your suggestions, which are more logical and give the reader a good reading experience. We then carefully amended the issue by rewriting the introduction. In the research context, we mainly emphasized the real-life development context of green finance ecosystems. Meanwhile, after sorting out that countries worldwide are facing green financial risks, we directly listed the research questions. The modifications are mainly in the introductory section on pages 2-3.

Comment 2. Add literature review. You can move some contents from the part of introduction but make sure to rewrite this part, because literature review in the original introduction part is just a list of reference with short depiction.

Response: Following your suggestions, we have also added the Literature Review section, which we divided into two subsections: green finance and green finance risk management. Each section summarizes the theoretical foundations and representative literature in detail.

On the one hand, we sorted out the research results of scholars at home and abroad on the theoretical mechanism of the green financial system and the driving mechanism of green financial development. We analyzed and explored the behavioral decision-making of green financial institutions with the help of the game model constructed by scholars among the government, banks and enterprises. The revisions are in the Green Finance section on pages 4-6.

On the other hand, with the help of the existing research results in the academic field, we summarize the literature on green financial risk management in terms of theoretical mechanisms and risk assessment methods. The paper modifications are in the in the Green Finance Risk Management section on page 6-7.

Comment 3. I don't believe "Differential Game Model" is a good section title. Please change it.

Response: In conjunction with your suggestions, we also felt that the original title could have been more inappropriate, and we have consequently revised the title of the section, which is now more clearly structured, highlighting the chapter's main research elements and increasing the paper's readability. The specific modifications are listed in the table below.

Comment 4. It seems that Model Assumption does not exactly match research questions.

Response: All of our authors carefully checked the Model Assumption. Meanwhile, we got help from an applied mathematics major who studies game theory. Now, the Model Assumptions are dramatically improved. Specifically: 

Hypothesis 1 is an assumption about the cost of inputs from the three subjects, which is used to calculate the HJB equation for each subject subsequently.

Hypothesis 2 is a dynamic assumption on the risk management level of the green financial ecosystem. The dynamic assumptions are made for the process of upgrading. This is used to calculate the HJB equation for each subject in the three contracts.

Hypothesis 3 assumes the total benefits of the green finance ecosystem at a given time, which is used to compute the objective function for three subjects.

Hypothesis 4 is an assumption about the objective function of the three main subjects based on the assumption of total benefits of the previous green financial ecosystems.

Comment 5. In Model Simulation, parameter values are set without reason. Do they realize the simulation of reality? It looks more like an experiment report.

Response: For the parameter assignment problem in the numerical simulation of differential game models, scholars usually base their assignment methods on practical experience or refer to existing research results. This is because the differential game model does not conduct experiments, and there is no accurate data obtained from experiments.

Our logic for assigning variables in a numerical simulation is combining the parameter assignment logic of scholars [30], referring to the indicator data on green finance in the National Bureau of Statistics and basing on the constraints of the model, the range of parameter values and the actual situation of the green financial market, given the parameter values, the results of the model will be compared using example analysis to verify the conclusions of the study.

The corresponding modifications are in the Model Simulation subsection on page 24.

Finally, once again, I would like to express my sincere thanks to you.

To Reviewer 2:

Thank you very much for your precious comments on this paper, which will improve it greatly. We have made serious revisions, and below are the specific revised responses.

Comment 1. There is a lack of in-depth exploration of specific risks and their mitigation strategies. A more detailed discussion would provide a holistic view of challenges in the green financial ecosystem, and a qualitative discussion of other approaches that have been introduced in the literature, using similar models, such as A Paradigm Shift Toward Satisfaction, Realism and Efficiency in Wireless Networks Resource Sharing, doi: 10.1109/MNET.011.2000368, is needed.

Response: Thank you for recommending the paper, which we read and were inspired accordingly. After the question was posed, we revised the realistic background of the research topic and sorted out the differences between green financial risks and traditional risks. On the one hand, green financial risks are highly uncertain. On the other hand, it is difficult to manage the risks affecting the implementation of green finance. In the meantime, we summarized the close connection between green financial development and green financial risk management, balancing future development opportunities with current innovation expansion and deepening. The modifications are mainly in the introductory section on pages 2-3.

In addition, we have combed through and summarized the literature you recommended in detail, uncovering the shortcomings of this paper. The modifications are the Research Gaps and Future Prospects section on page 32.

Comment 2. The paper introduces three contract modes but does not delve deeply into the theoretical mechanisms underlying each mode. A more thorough theoretical discussion would enhance the reader's understanding of the contractual dynamics.

Response: For this suggested revision, we mainly revised the theoretical mechanisms underlying each mode of the paper. We have carefully read the reference papers and have been incredibly enlightened.

First, the theoretical underpinning behind the no-incentive contract is the Nash equilibrium: any participating subject unilaterally changing his or her strategy under this combination of strategies (with no change in the strategies of the other participating subjects) will not improve his or her payoff. The revision is Non-incentive Contract section on pages 11-12.

Second, the theoretical underpinning behind the cost-sharing contract is the Stackelberg equilibrium: a framework in which the leader acts first, setting a strategy. Then, the followers determine their optimal response based on the leader's strategy. The revision is the Cost-sharing Contract on page 15.

Third, the theoretical support behind the collaborative cooperation contract: under this cooperation model, all the participating subjects, as an organic whole, aiming at maximizing the overall benefits, jointly negotiate to determine the optimal level of their respective input efforts as well as the optimal level of benefits for the system as a whole. The modifications are in the first paragraph of the Synergistic Contract section on page 19.

Comment 3. The introduction provides a qualitative overview of previous studies on green finance, but a more structured presentation, perhaps in the form of a literature review, could enhance the coherence and depth of the introduction.

Response: We are very grateful for your suggestions, especially the introduction part of the paper, which could not be more logical and give the reader a worse reading experience. We then carefully amended the issue by rewriting the introduction and adding a Literature Review section.

In the Introduction section, we mainly introduce the background of developing the reality of green finance and green financial risk management. Meanwhile, after sorting out the vacancies in the research field, we directly put forward the research questions.

In the Literature Review section, for the existing research results, we organize them in two subsections: green finance and green financial risk management.

The modifications are mainly in the introduction and literature review sections on pages 2-7.

Comment 4. The paper lacks details on the reproducibility of the numerical simulation analysis. Providing sufficient information on parameters and methodologies would enable other researchers to validate and build upon the results.

Response: Incorporating your and reviewer 1's suggestions, we have rewritten the Model Simulation section of the paper to focus on sufficient information on parameters and methodologies. The revisions are on page 24. The specific modifications are:

Our logic for assigning variables in a numerical simulation is combining the parameter assignment logic of scholars [30], referring to the indicator data on green finance in the National Bureau of Statistics and basing on the constraints of the model, the range of parameter values and the actual situation of the green financial market, given the parameter values, the results of the model will be compared using example analysis to verify the conclusions of the study.

Comment 5. Additionally, clearer visual aids or tables summarizing key numerical results would improve the presentation's clarity.

Response: Thank you for recommending the paper, which we read and were inspired accordingly. The parameters in the pre-revised paper were unintuitive and did not convey information such as model use.

The original parametric description of the paper was “Suppose the parameter values are as follows: , , , , , , , , , , , , , , , ”.

The corrected parametric description of the paper is shown in the table below.

Comment 6. The conclusion briefly mentions the contributions but could elaborate on potential avenues for future research. Suggestions for further investigations or extensions of the current model would enhance the paper's completeness.

Response: We have rewritten the Research Gaps and Future Prospects of the paper in such a way that, first, we emphasize some limitations of this paper, then we look forward to further research extensions of the current model to enhance the completeness of this field in the future.

The revised Research Gaps and Future Prospects for the paper is:

This paper divides the participants in the green financial ecosystem into three types of subjects. However, the same subject within the inevitable also involves a more complex competitive relationship. In order to simplify the model, this paper does not model analysis of this problem. Therefore, the complex green financial ecosystem risk management of the problem of cooperation needs to be further studied. In addition to the incentives studied in this paper, there are factors like green goodwill [36], green business process maturity [37], and green startup maturity [38] that affect the level of effort of the participants. In addition, with the increase of green finance user data and the number of green finance products, building the optimal structural composition of the green finance risk management system so that the participating subjects can continue to create value in a stable state also needs to be explored in depth. Future research could focus on such issues and try to study them from the perspectives of more complex competing relationships, multiple incentives, etc. In addition, the optimal structure and subject relationship of risk management in the green financial ecosystem can be studied so that the innovators can continue to promote high-quality risk management in a stable state and thus create more value.

Finally, once again, I would like to express my sincere thanks to you.

---

## [Decision Letter · Decision Letter 1]

1 Mar 2024

Research on Risk Management Incentive Strategy Based on the Green Financial Ecosystem

PONE-D-23-37731R1

Dear Dr. Lu,

We’re pleased to inform you that your manuscript has been judged scientifically suitable for publication and will be formally accepted for publication once it meets all outstanding technical requirements.

Kind regards,

Xin Shen

Academic Editor

PLOS ONE

Additional Editor Comments (optional):

Reviewers' comments:

Reviewer's Responses to Questions

**Comments to the Author**

1. If the authors have adequately addressed your comments raised in a previous round of review and you feel that this manuscript is now acceptable for publication, you may indicate that here to bypass the “Comments to the Author” section, enter your conflict of interest statement in the “Confidential to Editor” section, and submit your "Accept" recommendation.

Reviewer #1: All comments have been addressed

Reviewer #2: All comments have been addressed

2. Is the manuscript technically sound, and do the data support the conclusions?

Reviewer #1: Yes

Reviewer #2: Yes

3. Has the statistical analysis been performed appropriately and rigorously? 

Reviewer #1: Yes

Reviewer #2: Yes

4. Have the authors made all data underlying the findings in their manuscript fully available?

Reviewer #1: Yes

Reviewer #2: Yes

5. Is the manuscript presented in an intelligible fashion and written in standard English?

Reviewer #1: Yes

Reviewer #2: Yes

6. Review Comments to the Author

Reviewer #1: Thank you for the revision. Authors have addressed my comments raised in the previous review, although some parts still need to be improved, and I feel that this manuscript is now acceptable for publication.

Reviewer #2: All comments have been addressed thoroughly by the authors. I have no further concerns about this manuscript.

7. PLOS authors have the option to publish the peer review history of their article (what does this mean?). If published, this will include your full peer review and any attached files.

Reviewer #1: No

Reviewer #2: No

---

## [Editor Report · Acceptance letter]

13 Mar 2024

PONE-D-23-37731R1 

PLOS ONE

Dear Dr. Lu, 

I'm pleased to inform you that your manuscript has been deemed suitable for publication in PLOS ONE. Congratulations! Your manuscript is now being handed over to our production team.

Kind regards, 

on behalf of

Professor Xin Shen 

Academic Editor

PLOS ONE